# Self-attenuating adenovirus enables production of recombinant adeno-associated virus for high manufacturing yield without contamination

Weiheng Su[1,2], Maria I. Patrício [2], Margaret R. Duffy[1], Jakub M. Krakowiak [2], Leonard W. Seymour [1✉] & Ryan Cawood[2]

Recombinant adeno-associated virus (rAAV) shows great promise for gene therapy, however scalability, yield and quality remain significant issues. Here we describe an rAAV manufacturing strategy using a 'helper' adenovirus that self-inhibits its major late promoter (MLP) to truncate its own replication. Inserting a tetracycline repressor (TetR) binding site into the MLP and encoding the TetR under its transcriptional control allowed normal adenovirus replication in the presence of doxycycline but only genome amplification and early gene expression (the 'helper' functions) in its absence. Using this self-inhibiting adenovirus we demonstrate delivery of adenoviral helper functions, AAV rep and cap genes, and the rAAV genome to yield up to 30-fold more rAAV vectors compared to the helper-free plasmid approach and significant improvements in particle infectivity for a range of serotypes. This system allows significant improvements in the production of serotypes rAAV2, rAAV6, rAAV8 and rAAV9, and enables propagation of existing rAAV without transfection, a process that improves batch quality by depleting reverse packaged DNA contaminants. We propose this as a high-yielding, contaminant-free system suitable for scalable rAAV manufacture.

[1] Department of Oncology, University of Oxford, Old Road Campus, OX3 7DQ Oxford, UK. [2] OXGENE Ltd, Oxford Science Park, OX4 4HG Oxford, UK.
✉email: len.seymour@oncology.ox.ac.uk

Gene therapy strategies using recombinant adeno-associated virus (rAAV) are showing promise for several diseases[1]. rAAV is usually manufactured using 'helper-free' (HF) plasmid transfection of cells or with a 'helper' adenovirus[2–4]. The HF system is difficult to scale, while the helper approach requires the removal of contaminating adenovirus[5–7]. Both issues significantly increase manufacturing costs. Here we present an improved transfection-free helper adenovirus system that overcomes these limitations.

The adenovirus life cycle involves early and late temporal phases. The early phase provides helper functionality for rAAV production, whilst the late phase is responsible for the production of adenoviral structural proteins[8–10]. These proteins are mainly transcribed from the adenovirus major late promoter (MLP)[11]. We hypothesized that transcriptional regulation of the MLP would allow the provision of helper functions for rAAV manufacture, but truncate the adenovirus life cycle, thereby preventing adenovirus contamination of an rAAV preparation.

We describe an engineered adenoviral vector where the MLP is modified to include a repressor binding site (tetracycline operator, TetO) and where the repressor (TetR) is encoded under MLP transcription. This creates a doxycycline-controllable negative feedback loop regulating the expression of the adenovirus structural genes. This 'Tetracycline-Enabled Self-Silencing Adenovirus' (TESSA) shows adenoviral genome replication in both the presence and absence of doxycycline, but tight structural gene repression in its absence. We also stably encode AAV rep and cap genes providing, for the first time, all the adenoviral helper functionality and AAV genes within a single helper adenovirus.

We show that using two TESSA vectors (one encoding AAV rep and cap and one carrying the rAAV genome) can improve the rAAV yield for a range of serotypes by 10- to 30-fold and these particles are 5- to 60-fold more infectious when compared to the HF system. Adenoviral contamination levels were reduced by $1 \times 10^7$ fold compared to the use of normal helper adenovirus during rAAV production. Finally, we show that TESSA vectors encoding both AAV rep and cap genes allow the direct propagation of rAAV particles, and that this approach can reduce the level of DNA contaminants within an existing rAAV preparation.

## Results

**Engineering Tet-response elements into the adenovirus MLP.** We first identified MLP regions that allow insertion of TetO sites whilst maintaining wild-type MLP basal transcriptional activity[11]. We modified MLP reporter plasmids (pMLP-WT-EGFP) by introducing TetO sites immediately upstream, downstream, or flanking the MLP TATA box (Supplementary Fig. 1a). Modifications directly replaced existing MLP bases for those comprising the TetO site, where possible using synonymous codons and avoiding the DNA-binding and proof-reading domains of the adenoviral DNA polymerase on the opposite strand[12–14]. The modified DNA polymerase sequences are shown in Supplementary Fig. 1b[15]. MLP-TetO plasmids expressed EGFP levels comparable to wildtype-MLP (Supplementary Fig. 1c). By co-transfecting with plasmid pCMV-TetR in HEK293 cells and treating with doxycycline (or DMSO control) we found that EGFP expression could be regulated using a TetR binding element downstream of the MLP TATA box (Supplementary Fig. 1d).

**Doxycycline-controlled adenovirus that self-represses its own MLP.** We modified an E1/E3-deleted serotype 5 adenovirus vector (Ad5) by incorporating the chosen ('TetO1b') MLP sequence. This adenovirus showed normal replication kinetics, but supplying the TetR protein in trans could not fully suppress

adenovirus replication (Supplementary Fig. 1e). We, therefore, placed the TetR coding sequence under a splice acceptor directly under MLP transcriptional control, to create an auto-regulatory feedback to suppress MLP expression proportionally to adenovirus genome replication and MLP activation (termed TESSA).

We initially assessed the ability of the encoded TetR to auto-repress TESSA viral replication. HEK293 cells were infected with TESSA or the control E1/E3-deleted Ad5 (at a multiplicity of infection (MOI) of 5) and treated with DMSO or doxycycline. Infectious adenovirus was quantified by 50% cell culture infectious dose (TCID50) assay using the crude lysates harvested at day 3 post-infection. Representative brightfield microscopy images of the TCID50 assay taken at day 9 post-infection are provided in Supplementary Fig. 2a. Notably, viral plaques were observed in all wells (10/10) infected at 1 in $1 \times 10^6$ volume dilution with the control Ad5 or TESSA produced from HEK293 cells with doxycycline, confirming that TESSA yielded equivalent infectious titre (>1e8 TCID50/mL) compared to the control Ad5 (Supplementary Fig. 2b). In contrast, TESSA samples without doxycycline showed plaques in only ~2–3 out of 10 wells even at a much lower 1 in 10 volume dilution of the crude lysate, corresponding to fewer than 10 TCID50 per mL of infectious vectors and representing 99.99992% reduction compared to the control Ad5. This virtual absence of infectious viruses indicates that the MLP-TetO1b structure, with MLP-controlled TetR, gives almost total repression of adenovirus production in the absence of doxycycline.

**TESSA inhibits expression of late adenovirus proteins and enables rAAV production.** We next assessed the effect of MLP repression on the expression of the adenovirus early genes E2A DNA binding protein (DBP) and E4Orf6, which are known to be important for AAV replication. Quantification of E2A DBP and E4Orf6 mRNA by RT-qPCR showed comparable levels of E2A DBP transcripts in HEK293 cells infected with TESSA or the control Ad5, in both DMSO and doxycycline-treated group (Supplementary Fig. 2c). In contrast a ~3-fold increase in E4Orf6 mRNA was observed from TESSA (DMSO-group) compared to the other groups (Supplementary Fig. 2d). The reason for this remains unknown although it is conceivable that MLP repression somehow increases the production of E4Orf6, as increased expression of adenovirus early genes was previously reported from virus mutants deficient in transcription from the MLP[16].

We then assessed the ability of TESSA to deliver the requisite helper functions to produce rAAV vectors in HEK293 cells. We infected HEK293 cells, transfected with pRepCap2 and pAAV-EGFP (rAAV genome encoding EGFP), using TESSA or the control Ad5 (MOI of 10) and showed that DNAse-resistant rAAV2-EGFP yields (~$1 \times 10^9$ genome copies (GC) per mL) were comparable to the HF method (Supplementary Fig. 2e). This suggests that TESSA can provide all the essential helper functions. We also assessed the potency of the rAAV2-EGFP particles produced by transducing HEK293 cells. Crude preparations of rAAV2-EGFP were heat-treated at 60 °C for 30 min to inactivate infectious adenovirus that might influence transduction, and HEK293 cells were infected at a normalised 100 GC of rAAV per cell[17]. rAAV2-EGFP produced using the TESSA helper were equally potent compared to HF-derived vectors with >60% of cells positive for EGFP expression, while a slight reduction in infectivity was observed from the control Ad5 samples (Supplementary Fig. 2f).

For efficient rAAV production, we also inserted an rAAV genome encoding EGFP into the E1-deleted region of the TESSA to generate TESSA-AAV (Fig. 1a). This hybrid virus was designed to exploit the extensive genome replication of adenovirus for

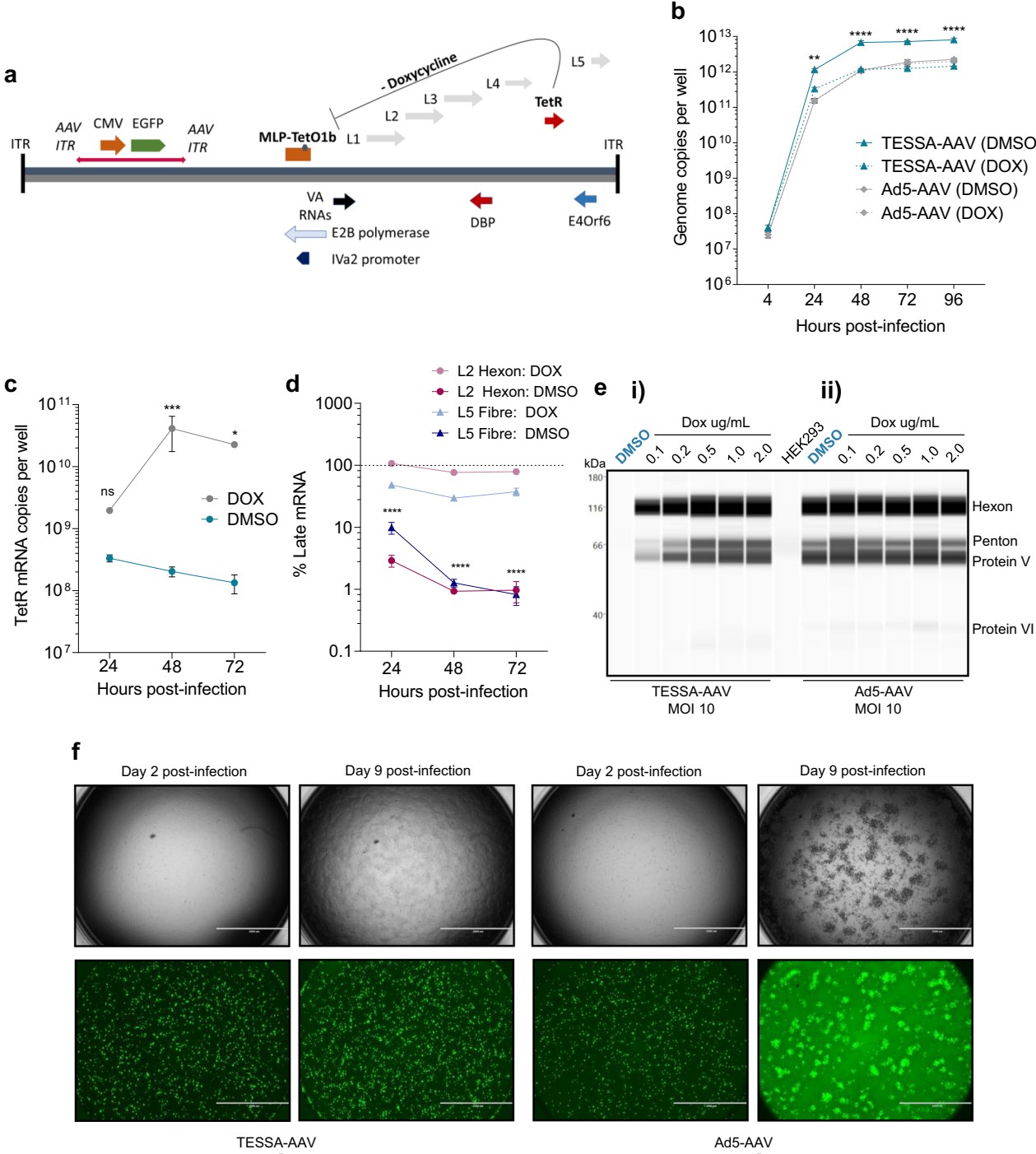

**Fig. 1 TESSA-AAV represses late adenoviral proteins and virus spread. a** Schematic of TESSA-AAV encoding an rAAV genome with a EGFP reporter in the E1-deleted region. **b** Comparison of genome replication, determined by fibre-specific qPCR, between TESSA-AAV and Ad5-AAV in HEK293 cells (MOI of 100) treated with doxycycline or DMSO. Data are $N = 3$ (mean ± SD) of biological replicates. Analysed by two-way ANOVA followed by Bonferroni post hoc test comparing TESSA-AAV (DMSO) versus Ad5-AAV (DMSO). **$p = 0.0019$, ****$p \leq 0.0001$. **c** Expression of TetR mRNA, as determined by RT-qPCR, from HEK293 cells infected with TESSA-AAV (MOI of 10) and treated with doxycycline or DMSO. Data are $N = 3$ (mean ± SD) of biological replicates. Analysed by two-way ANOVA followed by Bonferroni post hoc test comparing DMSO versus DOX group. *$p = 0.0440$, ***$p = 0.0007$. **d** Expression of Hexon and fibre mRNA from TESSA-AAV (MOI of 10) in HEK293 cells treated with doxycycline or DMSO, as determined by RT-qPCR. Data are shown as the percentage of mRNA levels compared to the control Ad5-AAV. Data are $N = 3$ (mean ± SD) of biological replicates. Analysed by two-way ANOVA followed by Bonferroni post hoc test comparing L2 Hexon and L5 Fibre DMSO group against the DOX group. ****$p \leq 0.0001$. **e** Western blot detection of adenovirus structural proteins from HEK293 cells infected with TESSA-AAV (**i**) or Ad5-AAV (**ii**) used at an MOI of 10, and treated with DMSO or escalating doses of doxycycline. **f** Representative images of HEK293 cells infected at a $1 \times 10^{-7}$ volume dilution with TESSA-AAV versus Ad5-AAV. Cells were imaged at day 2 and day 9 post-infection. Scale bar, 1000 μm. For panel **e** and **f**, data representative of at least three independent experiments.

amplification of the integrated rAAV genome sequence, allowing improved rAAV rescue and subsequent rAAV genome replication[13,18]. We also constructed a control Ad5 encoding the rAAV-EGFP genome (Ad5-AAV).

Genome replication of Ad5-AAV in HEK293 cells, determined by fibre-specific qPCR, was unaffected by doxycycline, and in its presence, TESSA-AAV exhibited similar replication kinetics as the Ad5-AAV (Fig. 1b). During scale-up of TESSA-AAV production, in the presence of doxycycline, no differences were observed in virus yield and quality (Supplementary Fig. 3a), suggesting that adenovirus infectivity and replication is unaffected by the TESSA modifications. In contrast, in the absence of doxycycline, no TESSA-AAV adenovirus particles could be banded (Supplementary Fig. 3b) although qPCR analysis of cell lysate showed that TESSA-AAV produced ~10-fold ($p \leq 0.0001$) more adenovirus genomes than Ad5-AAV (Fig. 1b).

To demonstrate the ability of TESSA-AAV to transcribe TetR and auto-repress its MLP-TetO1b, we showed high expression of TetR mRNA in doxycycline-treated HEK293 cells at 24 h post virus infection (hpi; Fig. 1c) and increased by ~20-fold at 48 hpi. This suggested transcription of TetR from the virus, coupled with functional inhibition of the TetR protein by doxycycline to prevent repression of the MLP-TetO1b. In contrast, DMSO-treated cells showed over 200-fold fewer TetR mRNA transcripts at 48 hpi, confirming that MLP-transcribed TetR can inhibit further transcription from MLP-TetO1b (Fig. 1c). TetR protein expression from the TESSA vector shows a similar pattern by western blot analysis (Supplementary Fig. 3c). We also quantified mRNAs encoding adenoviral structural proteins Fibre and Hexon. In DMSO-treated cells, TESSA-AAV shows 80-fold lower hexon and 46-fold lower fibre mRNA at 72 hpi compared to treatment with doxycycline (Fig. 1d). Structural proteins from TESSA-AAV were also fully suppressed in the absence of doxycycline (Fig. 1e (i), first lane) compared to Ad5-AAV (Fig. 1e (ii), first lane) which showed high levels of adenovirus capsid proteins. Repression of TESSA-AAV viral replication can be easily visualised in HEK293 cells as it encodes the EGFP reporter. As shown in Fig. 1f, cells infected with TESSA-AAV or Ad5-AAV were positive for EGFP expression and can be notably observed at day 2 post-infection. However, while the Ad5-AAV infection eventually spread across the well at day 9 with advanced viral cytopathic effect (CPE) and cell lysis, the spread of TESSA-AAV was inhibited and remained confined to cells from the initial inoculation, confirming that self-repression of MLP-TetO1b from TESSA-AAV remained intact throughout. Strong repression of the TESSA MLP persists at MOI up to 1000 (Supplementary Fig. 3d).

We also assessed rescue and replication of the rAAV-EGFP genome from TESSA-AAV for rAAV production. HEK293 cells transfected with pRepCap2 and infected with TESSA-AAV or the control Ad5-AAV resulted in ~8-fold increase in DNAse-resistant rAAV2 vectors (~$8 \times 10^9$ GC/mL) compared to the HF group (Supplementary Fig. 4a). As expected, high levels of contaminating Ad5 (~$3 \times 10^9$ GC/mL) were detected by hexon-specific qPCR from the control Ad5-AAV which was absent from the TESSA-derived and HF-derived samples. The addition of crude rAAV2 preparations produced from Ad5-AAV or TESSA-AAV (doxycycline-treated group) to fresh HEK293 cells results in early and high levels of EGFP notable at 24 hpi, thought to reflect adenovirus-encoded EGFP, and advanced viral CPE at 96 hpi. In contrast, crude rAAV2 produced from TESSA-AAV without doxycycline showed delayed EGFP expression, typical of the expression kinetics of rAAV due to the requirement for genome second-strand synthesis[19], and an intact cell monolayer absent of focal spreading of EGFP that is characteristic of adenovirus vector contamination (Supplementary Fig. 4b).

**A TESSA system encoding all adenovirus helper functions plus AAV rep and cap.** While rAAV genomes can be stably encoded within the E1-deleted region of adenovirus[18,20] and TESSA (Fig. 1b) without affecting adenovirus replication, encoding AAV rep and cap is challenging because the Rep coding sequences and proteins are both toxic to adenovirus replication[21–24].

AAV Rep proteins repress the early (E1a, E2a and E4) and late (MLP) adenovirus promoters[25,26], while the AAV p40 promoter region is a potent inhibitor of adenovirus replication *in cis*[22]. As a first step, the cap gene from AAV serotype 2 was inserted in reverse orientation into the E1-deleted region of TESSA, under CMV promoter control. To stably incorporate AAV rep we adopted a rational combinatorial approach to alleviate its toxicities to adenovirus. First, to enable transcriptional control of the primary rep genes (rep78/68) and reduce expression of the smaller rep52/40 isoforms that can also limit adenovirus replication, we attempted to ablate the AAV p19 promoter by synonymous codon exchange of the TATA box[27]. Secondly, the AAV p40 promoter region[21,22] was scrambled by synonymous codon exchange to maintain rep78/68 coding frames and reduce its adenovirus toxicity. Finally, because minimising Rep78/68 expression can enhance rAAV production[28,29] we incorporated rep78/68 without any promoter within the E1-deleted region, in the same orientation as the cap gene[30], and positioned at a distance away from the adenovirus left ITR sequence to avoid transcriptional activity that could result in increased expression of Rep[31]. We term this adenovirus TESSA-RepCap2 (Supplementary Fig. 4c). Surprisingly, despite attempts to suppress the expression of Rep52 and Rep40 by ablation of the p19 TATA box, we found that all four Rep isoforms were readily detected with Rep52 and Rep40 expressed in excess compared to Rep78/68 after infection of HEK293 cells using TESSA-RepCap2 (Supplementary Fig. 4d). Rep expression from TESSA was significantly higher than in cells transfected with pRepCap2, but expression patterns of the Rep isoforms were comparable between the two groups. Whilst the high expression of Rep52 and Rep40 following p19 TATA box ablation was unexpected, subsequent assessment of the literature identified similar findings[32]. This was particularly encouraging as Rep52 and Rep40 have been suggested as essential for rAAV production and to be involved in DNA encapsidation via helicase activity[33].

**Use of TESSA2.0 system for the manufacture of rAAV2.** To assess the amount and quality of rAAV2 produced using TESSA, adherent HEK293 cells were co-infected with TESSA-RepCap2 and TESSA-AAV (together 'TESSA2.0'), each at an MOI of 25, and the results compared with the HF approach. Levels of DNAse-resistant rAAV2 were determined using an EGFP-specific qPCR assay. The HF system produced ~$1–2 \times 10^4$ GC/cell of rAAV2 at 48- and 72 h post-treatment (hpt), consistent with previous reports[34–36]. However, we also observed a slight decline in productivity at 96 hpt from the HF method. At 48 hpt, TESSA2.0 yielded over $2 \times 10^5$ GC/cell of rAAV2, which increased to >$3 \times 10^5$ GC/cell at 96 hpt (Fig. 2a). This represented an average of over 20-fold greater rAAV2 vector yield compared to the HF method. These results were confirmed by an AAV2 specific ELISA directed against assembled capsids which showed over $6 \times 10^{11}$ viral particles (VP)/mL from TESSA2.0 compared to ~$8 \times 10^{10}$ VP/mL from the HF method (Supplementary Fig. 5a). Assessment of encapsulated rAAV2 genomes (% full) determined by comparing viral particles against the qPCR titre showed a twofold higher proportion of full rAAV2 particles (~10% containing the EGFP transgene) from TESSA2.0 compared to rAAV2 derived from HF (Supplementary Fig. 5b). Similarly, a 2-fold greater proportion of full capsids was observed

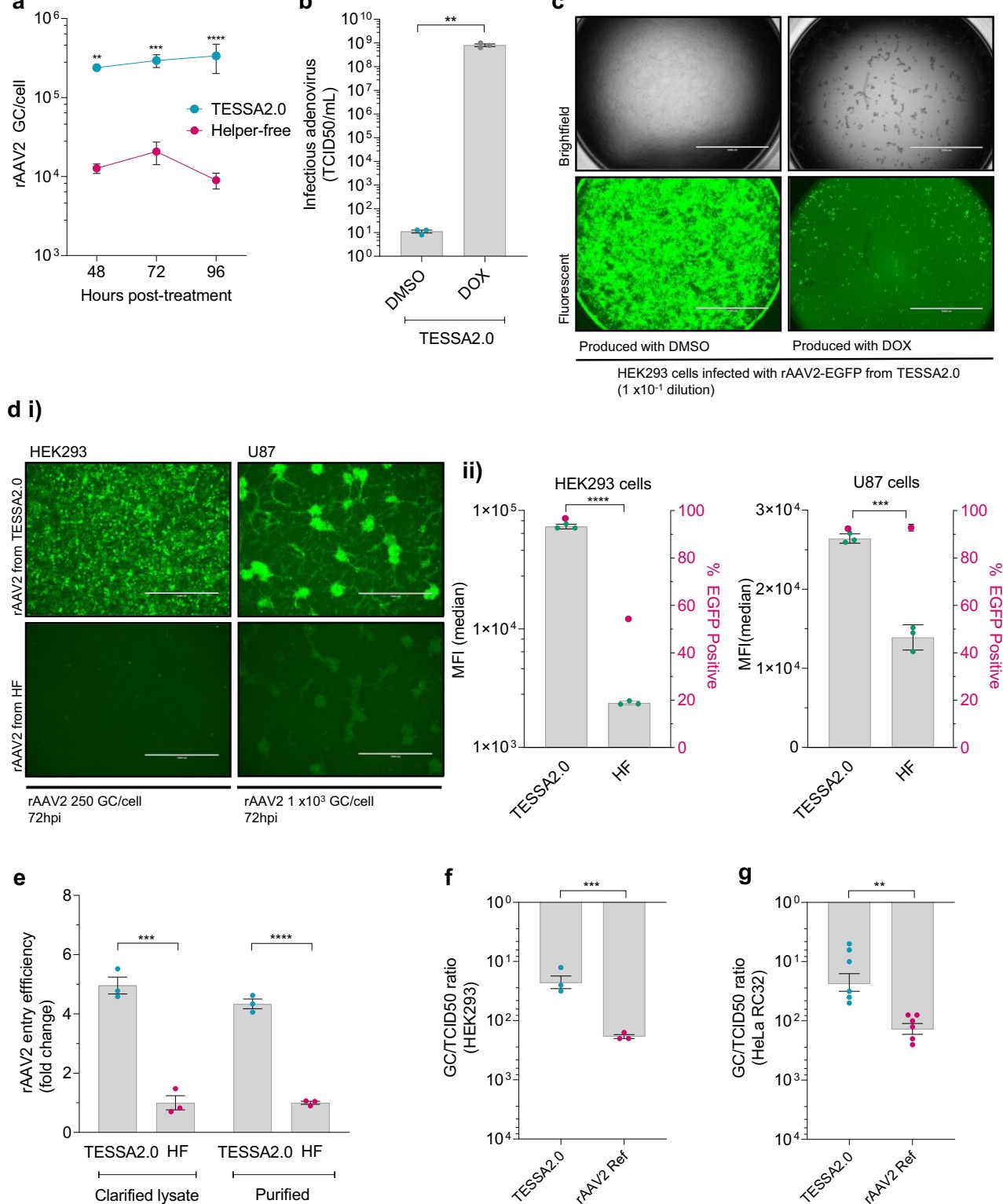

in purified stocks (via AAVX Affinity Resin) of rAAV2 produced in suspension HEK293 cells using TESSA2.0 compared to a commercial rAAV2 reference standard (Supplementary Fig. 5c).

Analysis of TESSA2.0-derived rAAV2 by transmission electron microscopy (TEM) showed the expected AAV morphology, indistinguishable from HF-derived particles, and no evidence for any adenovirus contamination (Supplementary Fig. 5d). We also measured adenovirus contamination in crude rAAV preparations

by quantifying infectious particles using a TCID50 assay in the presence of doxycycline. When doxycycline was present during manufacture, TESSA went through the full adenoviral life cycle, giving $8 \times 10^8$ TCID50/mL of infectious adenovirus. However, in the absence of doxycycline, the conditions used for rAAV manufacture, contamination fell by almost $10^8$-fold to just ~10 TCID50/mL (Fig. 2b). This almost complete absence of contaminating adenovirus was also confirmed by hexon-specific

**Fig. 2 TESSA2.0 enhances rAAV2 production in HEK293 cells. a** Production of rAAV2-EGFP in HEK293 cells using the TESSA2.0 vectors (TESSA-AAV and TESSA-RepCap2 used at an MOI of 25) versus transfection with the helper-free (HF) plasmids. DNAse-resistant particles quantified by EGFP-specific qPCR. Data are $N = 3$ (mean ± SD) biological replicates. Analysed by two-way ANOVA followed by Bonferroni post hoc test comparing TESSA2.0 versus Helper-free. **$p = 0.0018$, ***$p = 0.0004$, ****$p \leq 0.0001$. **b** TCID50 assay quantification of infectious adenovirus in crude rAAV2-EGFP preparations generated from the TESSA2.0 approach (DMSO-treated group) compared to treatment with doxycycline. Data are $N = 3$ (mean ± SEM) biological replicates. Statistical significance was calculated using a two-tailed unpaired $t$-test. **$p = 0.0016$. **c** Representative brightfield and fluorescence microscopy images of HEK293 cells infected at a 1:10 volume dilution with rAAV2-EGFP crude preparations derived from TESSA2.0 that were produced in presence of DMSO or doxycycline. Cells supplemented with doxycycline and imaged at day 7 post-infection. Scale bar, 2000 μm. Data representative of at least three independent experiments. (**d i**) Fluorescent images of HEK293 and U87 cells transduced with crude preparations of rAAV2-EGFP derived from the TESSA2.0 or HF approach. Cells were imaged at 72 hpi (Scale bar, 1000 μm) and EGFP expression analysed by (**ii**) flow cytometry. Data are $N = 3$ (mean ± SD) biological replicates. Statistical significance was calculated using a two-tailed unpaired $t$-test. ***$p = 0.0002$, ****$p \leq 0.0001$. **e** Efficiency of cellular uptake of rAAV2-EGFP particles derived from TESSA2.0 compared to HF. HEK293 cells were infected at 50 GC/cell with clarified lysate or purified preparation of rAAV2-EGFP. Total genomes were quantified by EGFP-specific qPCR at 6 hpi and data presented as fold-change relative to cells infected with rAAV2-EGFP derived from HF. Data are $N = 3$ (mean ± SEM) biological replicates. Statistical significance was calculated using a two-tailed unpaired $t$-test. ***$p = 0.0004$, ****$p \leq 0.0001$. Transduction potency of rAAV2-EGFP derived from TESSA2.0 compared to a control rAAV2-EGFP reference stock (Vector BioLabs). Purified DNAse-resistant rAAV2-EGFP stocks were normalised by EGFP-specific qPCR to equivalent titres and transduction quantified by the TCID50 assay in (**f**) HEK293 cells and (**g**) HeLa RC32 cells aided by wt Ad5. Data presented as GC to TCID50 ratio. For panel **f**, data are $N = 3$ (mean ± SEM) biological replicates. For panel **g**, $N = 6$ (mean ± SEM) biological replicates from two independent experiments. Statistical significance was calculated using an unpaired $t$-test (two-tailed). **$p = 0.0032$, ***$p = 0.0004$.

qPCR (Supplementary Fig. 5e). The drastic reduction in contaminating adenoviruses was notable from infecting new HEK293 cells with crude rAAV2-EGFP preparations generated using TESSA2.0 (± doxycycline supplement). HEK293 cells (with doxycycline supplement) infected with rAAV-EGFP from the TESSA2.0 DMSO-group showed significant expression of EGFP from the rAAV particles, but free of adenovirus plaques and CPE, while advanced CPE and cell death was observed from the TESSA2.0 doxycycline-group (Fig. 2c). Importantly, any low level of adenovirus contamination is expected to retain the double TESSA attenuation, and the absence of plaques observed in the absence of doxycycline is commensurate with zero replication-competent adenovirus contamination.

We assessed the infectivity of these rAAV2 preparations by adding crude preparations from TESSA2.0 or HF to new cells and monitoring EGFP expression. At equal numbers of viral vectors, rAAV2 produced by TESSA2.0 showed markedly greater transduction per particle than HF-derived rAAV2 in both HEK293 and U87 cells (Fig. 2d). The mechanism of improvement appeared to partly involve increased cell-entry, as a measurement of rAAV2 genomes per cell after incubation with equal numbers of rAAV2 (clarified lysate and purified vector stocks) made by TESSA2.0 and HF showed the former achieved 5-fold greater uptake (Fig. 2e). Transduction potency of purified rAAV2-EGFP produced from TESSA2.0 was compared to a commercial rAAV2 reference stock (produced by the HF method) in HEK293 cells using the TCID50 assay. TESSA2.0-derived rAAV particles exhibited a GC/TCID50 ratio of ~25, an eightfold greater transduction potency compared to the rAAV2 reference stock (GC/TCID50 ratio of 200 (Fig. 2f)). rAAV transduction was also compared in HeLa RC32 cells, stably encoding the AAV2 rep and cap genes, and in the presence of wildtype (wt) Ad5 infection. In these cells, TESSA2.0-derived particles exhibited a GC/TCID50 ratio of 23, a sixfold greater transduction potency compared to the rAAV2 reference stock (GC/TCID50 ratio of 140 (Fig. 2g)).

**Production of rAAV serotypes 6, 8 and 9 using TESSA2.0.** We then assessed the utility of the TESSA2.0 approach for production of rAAV serotypes 6, 8 and 9 using TESSA-RepCap6, -8 and -9, respectively, with TESSA-AAV encoding the transfer genome. Productivity of rAAV6 (Fig. 3a i), rAAV8 (Fig. 3b i), and rAAV9 (Fig. 3c i) were compared to the HF method using the corresponding pRepCap6, -8 and -9 plasmids. Significant increases in rAAV production were observed in all assessed serotypes, with

rAAV6 and rAAV8 yielding near $1 \times 10^6$ DNAse-resistant genomes per cell. Specifically, the TESSA2.0 approach yielded an average of 20-, 25- and 10-fold greater yields of rAAV6, 8 and 9 vectors, respectively, compared to the HF productions.

Assessment of the proportion of full capsids by comparing assembled particles against the qPCR titre shows that both rAAV6 (Fig. 3a ii) and rAAV8 (Fig. 3b ii) from TESSA2.0 contained a twofold higher proportion of full particles (~15-20% containing the EGFP transgene) compared to HF-derived samples. However, rAAV9 from TESSA2.0 and HF both showed ~6% full capsids (Fig. 3c ii).

As with rAAV2, when applied at equal vector numbers per cell, rAAV6 produced using TESSA2.0 showed markedly greater specific infectivity in both HEK293 and U87 cells compared to HF (Fig. 3a iii). Direct comparison of purified rAAV6, rAAV8 and rAAV9 stocks produced in suspension HEK293 cells from TESSA2.0 against commercial reference controls using a TCID50 assay in HeLa RC32 cells (with wt Ad5), showed that rAAV6, rAAV8 and rAAV9 from TESSA2.0 exhibited a GC to TCID50 ratio of 60, 15000, and 1600, respectively. These levels represented improved transduction potency over the commercial reference stock of 60-fold for rAAV6 (Fig. 3a iv), 7-fold for rAAV8 (Fig. 3b iii) and 25-fold for rAAV9 (Fig. 3c iii).

**Propagation of rAAV using TESSA-RepCap.** We next assessed whether TESSA-RepCap2 could propagate AAV particles directly. This would remove the need to encode the rAAV genome within TESSA and allow existing rAAV material to be propagated without transfection. Accordingly, we infected HEK293 cells with rAAV2 particles (50 GC/cell) that had been produced either by TESSA2.0 or by the HF system, with and without TESSA-RepCap2. Interestingly, while a single infection with rAAV2 produced by TESSA gave a low level of EGFP in a high frequency of cells, EGFP was almost undetectable when cells were infected with rAAV produced by the HF approach alone (Fig. 4a, top images). Nevertheless, when TESSA-RepCap2 was added to these cells almost every cell became EGFP positive (Fig. 4a, lower images). This suggests that at least one HF-derived rAAV genome must have entered these cells for transduction and act as a template for replication by TESSA-RepCap2, even though transgene expression was not detected until infection with the TESSA-RepCap2.

Assessment of productivity showed that rAAV2 from either source (i.e. made by TESSA or HF) in combination with TESSA-

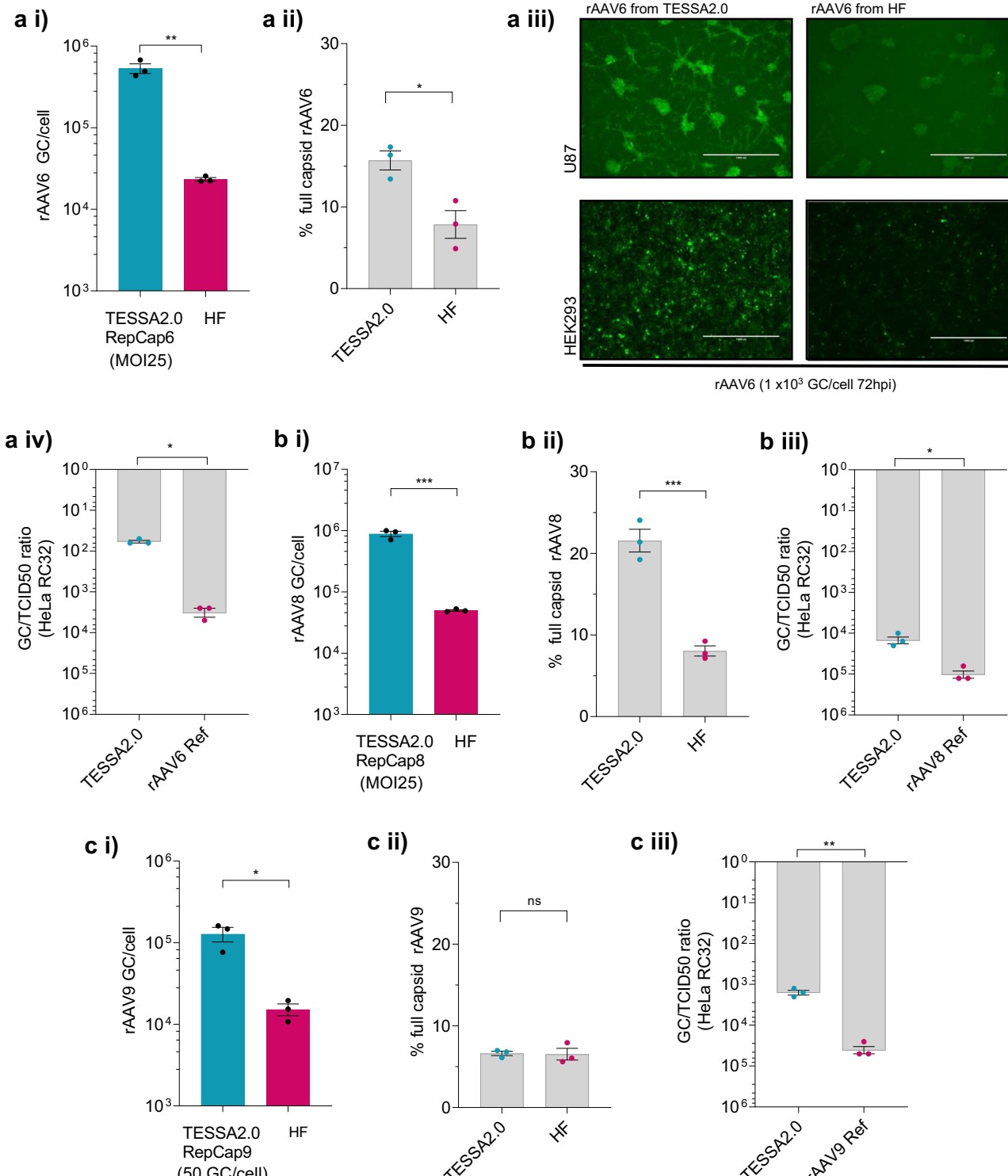

**Fig. 3 TESSA2.0 enhances production of rAAV6, rAAV9 and rAAV8 compared to the HF approach.** Production yield of (**a i**) rAAV6-EGFP, (**b i**) rAAV8-EGFP, and (**c i**) rAAV9-EGFP in HEK293 cells using TESSA2.0 vectors compared to the HF method. Data presented as DNAse-resistant genomes determined by EGFP-qPCR. Proportion of encapsulated genomes (% full) of (**a ii**) rAAV6-EGFP, (**b ii**) rAAV8-EGFP, and (**c ii**) rAAV9-EGFP determined following quantification of assembled rAAV6, -8, -9 particles using ELISA and compared against the qPCR titre. (**a iii**) Assessment of rAAV6-EGFP transduction in HEK293 and U87 cells. Cells were infected (1000 GC/cell) with crude rAAV6-EGFP preparations from TESSA2.0 or HF and imaged by fluorescent microscopy at 72 hpi (representative of $N = 3$ biological replicates). Scale bar, 1000 µm. Transduction potency of (**a iv**) rAAV6-EGFP, (**b iii**) rAAV8-EGFP and (**c iii**) rAAV9-EGFP derived from TESSA2.0 compared to a control rAAV6-EGFP, rAAV8-EGFP and rAAV9-EGFP reference stock (Vector BioLabs), respectively. Purified DNAse-resistant rAAV stocks were normalised by EGFP-specific qPCR ($1 \times 10^9$ GC/mL) and transduction quantified by the TCID50 assay in HeLa RC32 cells aided by wt Ad5. Data presented as GC to TCID50 ratio. For all panels, data are $N = 3$ (mean ± SEM) biological replicates. Statistical significance was calculated using an unpaired $t$-test (two-tailed). *$p \le 0.05$, **$p \le 0.01$, ***$p \le 0.001$.

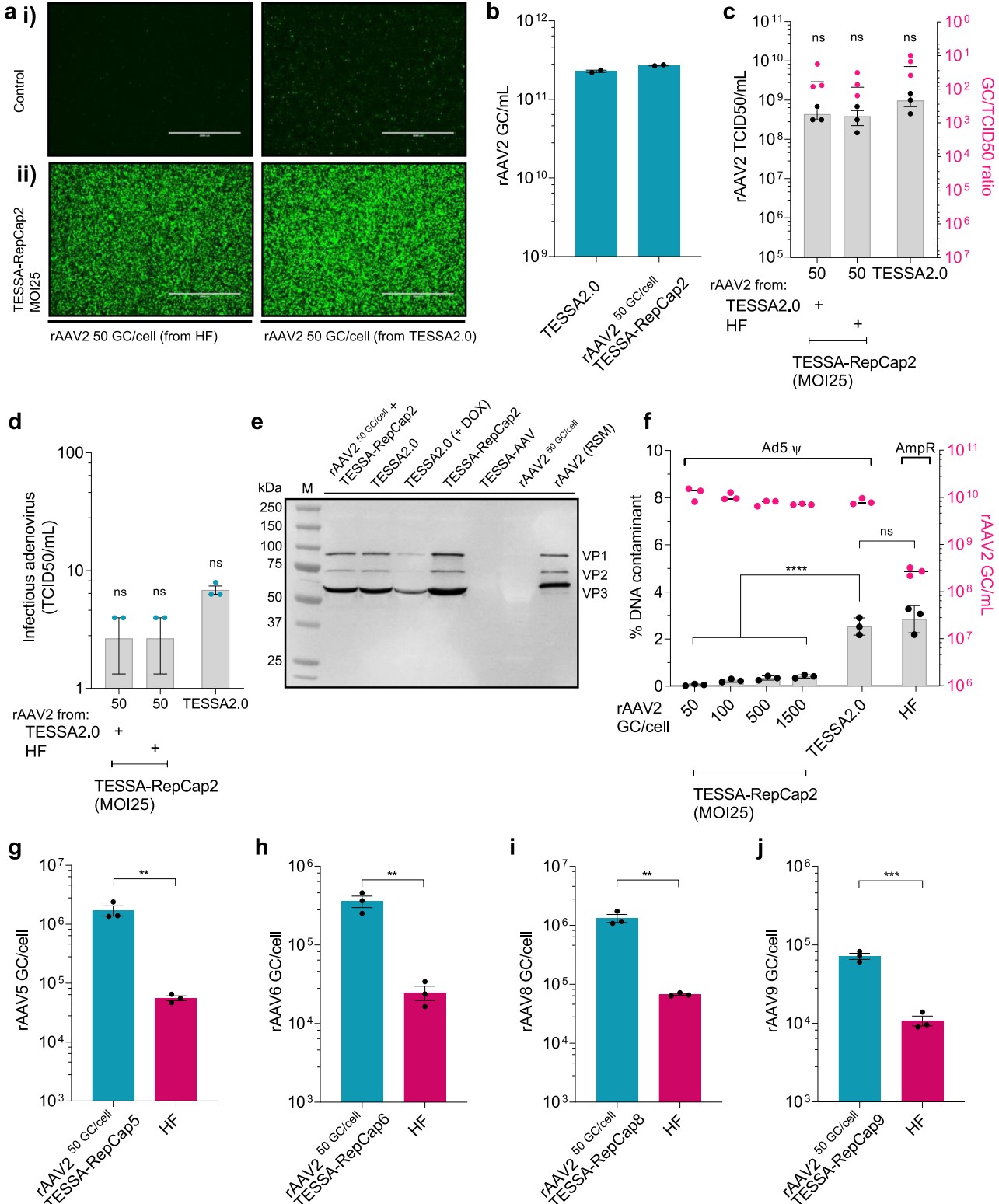

RepCap2 produced ~$1 \times 10^5$ GC/cell of rAAV2 by qPCR, comparable to yields from the TESSA2.0 approach (Supplementary Fig. 6a). We also assessed the potential to scale up manufacture using TESSA, and therefore tested infection of suspension HEK293 cells in a 1 L bioreactor. From this initial run, TESSA2.0 or co-infection of TESSA-RepCap2 plus rAAV2 yielded high amounts of DNAse-resistant rAAV2 vectors at ~$2.2 \times 10^{11}$ and $2.7 \times 10^{11}$ GC per mL of cell culture, respectively

(Fig. 4b). It, therefore, seems that these approaches may be suitable for large-scale rAAV manufacture. This production strategy could also be used for the production of a therapeutic rAAV vector. Infection of HEK293 cells with TESSA-RepCap2 alongside rAAV2 encoding human factor IX (hFIX) resulted in high vector yields of approximately >$5 \times 10^4$ GC/cell for rAAV2 encoding hFIX, including a self-complementing form (Supplementary Fig. 6b)[37,38].

**Fig. 4 Propagation of rAAV using TESSA-RepCap in HEK293 cells. a** Representative fluorescent images of HEK293 cells infected with (**i**) rAAV2-EGFP (50 GC/cell, control) or (**ii**) rAAV2-EGFP with TESSA-RepCap2 (MOI of 25). Imaged at 48 hpi. Scale bar, 2000 μm. Data representative of at least three independent experiments. **b** Production of rAAV2-EGFP using TESSA in 1 L stir-tank bioreactor culture of suspension HEK293 cells. Cells were co-infected with the TESSA2.0 vectors (TESSA-AAV and TESSA-RepCap2, each at an MOI of 25) or TESSA-RepCap2 (MOI of 25) co-infected with rAAV2-EGFP (HF-derived), at 50 GC/cell. DNAse-resistant genomes were quantified by EGFP-specific qPCR at 96 hpi and presented as GC per mL of cell culture. Data are $N = 2$ (mean ± SD) technical measurements. **c** Transduction potency of rAAV2 (produced using the TESSA2.0 vectors, each at an MOI of 25, or co-infection of rAAV2-EGFP at 50 GC/cell with TESSA-RepCap2) as determined in HEK293 cells using a TCID50 assay. GC/TCID50 ratio shown on the right axis was determined by comparing against the EGFP-specific qPCR titre of DNAse-resistant rAAV particles. Data are $N = 3$ (mean ± SEM) biological replicates (ns, analysed by one-way ANOVA with Tukey's multiple comparisons). **d** Assessment of adenovirus contamination in crude rAAV2 preparations derived from TESSA2.0, or via passage of rAAV2 using TESSA-RepCap2, using a TCID50 assay. Data are $N = 3$ (mean ± SEM) biological replicates. Analysed by one-way ANOVA with Tukey's multiple comparisons. **e** Western blot of AAV2 capsid proteins from particles produced in HEK293 cells using TESSA2.0 (± doxycycline), or via co-infection of rAAV2 particles (produced from TESSA2.0) at 50 GC/cell with TESSA-RepCap2 (MOI of 25). AAV2 reference standard material (RSM, ATCC VR-1616) is also shown. Data representative of at least three independent experiments. **f** Assessment of DNA contaminants in rAAV2 stocks produced via TESSA2.0, HF, or co-infection of TESSA-RepCap2 (MOI of 25) with escalating rAAV2 vector doses (produced using TESSA2.0). Impurities were determined in DNAse-resistant rAAV2 by qPCR against the adenovirus packaging signal Ψ (TESSA2.0-derived rAAV) or AmpR (HF-derived rAAV). Data are the percentage of adenovirus or AmpR DNA compared to EGFP. Titre of rAAV2 is shown on the right axis. Data presented are $N = 3$ (mean ± SD) biological replicates. Statistical significance was calculated by one-way ANOVA with Tukey's multiple comparisons. ****$p ≤ 0.0001$. Production yield of (**g**) rAAV5 (**h**) rAAV6 (**i**) rAAV8 (**j**) rAAV9 encoding EGFP from HEK293 cells co-infected with rAAV2 vectors (50 GC/cell) and TESSA-RepCap5 (100 GC/cell), TESSA-RepCap6 (100 GC/cell), TESSA-RepCap8 (100 GC/cell), or TESSA-RepCap9 (75 GC/cell), respectively. DNAse-resistant particles quantified by EGFP-specific qPCR at 96 hpi and compared to HF approach. Data presented are $N = 3$ (mean ± SEM) biological replicates. Statistical significance was calculated using an unpaired t-test (two-tailed). **$p ≤ 0.01$, ***$p = 0.0007$.

Infectious activity of rAAV2-EGFP produced from the propagation of rAAV stocks using TESSA-RepCap2 showed comparable potency to the TESSA2.0-derived particles when assessed visually (Supplementary Fig. 6c). This was further confirmed using the TCID50 assay (Fig. 4c). The greatest yields of infectious rAAV2 were obtained with TESSA2.0 ($9.7 × 10^8$ TCID50/mL), whereas rAAV2 particles used in conjunction with TESSA-RepCap2 produced slightly less transduction-competent rAAV2 ($4 × 10^8$ TCID50/mL). The level of contaminating adenovirus (measured by TCID50 in the presence of doxycycline) was very low in each preparation, and when rAAV2 was used as a starting material only two infectious adenoviruses were detected per mL (Fig. 4d).

Western blot analysis showed that expression of capsid proteins VP1, VP2 and VP3 were similar to that observed in ATCC-standard rAAV particles, showing ~1:1:10 ratio in particles made using TESSA2.0, TESSA-RepCap2 and also for cells infected with TESSA-RepCap2 alongside rAAV2 (Fig. 4e). In the presence of doxycycline, when the MLP is active, TESSA2.0 produced significantly less VP1, VP2 and VP3.

During ITR replication read-through to adjacent DNA can occur, and if this DNA is packaged it is often termed 'reverse packaging'[39]. We, therefore, measured the incorporation of adenovirus DNA into AAV particles produced using TESSA2.0. The qPCR assay against the adenovirus packaging signal (~325 bases from the left AAV ITR) showed that ~2.5% of AAV particles contain the adenovirus packaging signal (Fig. 4f). This compares to 2.8% of the AAV particles containing the ampicillin resistance gene (AmpR) using the HF approach, both within the expected reverse packaging range of 1–6%[39]. Since the adenovirus packaging signal DNA would not contain both terminal AAV ITR sequences, and would likely not replicate efficiently, we determined if the passage of rAAV2 in conjunction with TESSA-RepCap2 could reduce this contamination level. After a single passage of rAAV2 that was originally produced using TESSA2.0, the contamination level was decreased from 2.5 to 0.05% (Fig. 4f).

Finally, the propagation approach can be used to switch the serotype of rAAV vectors during manufacture. This is useful when the required rAAV serotype does not infect the packaging cells efficiently. For example, rAAV5, 6 and 9 are less efficient at infecting HEK293 cells than rAAV2 (Supplementary Fig. 6d)[40]. Accordingly, when HEK293 cells were infected with rAAV2-EGFP alongside TESSA-RepCap5, TESSA-RepCap6, or TESSA-

RepCap9 (Supplementary Fig. 6e), virtually all cells were successfully transduced with high expression of EGFP from replication of the rAAV-EGFP genomes by the TESSA-RepCap. Production yields of rAAV6 and rAAV9 produced using the capsid-exchange strategy were ~$1 × 10^5$ GC/cell, with rAAV5 and rAAV8 yielding above $1 × 10^6$ GC/cell and significantly above the HF production (Fig. 4g–j). By western blotting, the virus preparations maintained characteristic ratios of VP1-3 (Supplementary Fig. 7a).

To assess genetic stability, we serially passaged TESSA-RepCap2 to passage 7 and validated the stability of the virus genome by qPCR directed against the AAV and adenovirus genes. In each purified stock of TESSA-RepCap2 at various passages, AAV2 rep and cap DNA were found to be at equal copy numbers relative to the adenovirus hexon gene, indicating stable propagation of the AAV genes within TESSA-RepCap2 (Supplementary Fig. 7b). Genetic stability was also validated at each serial passage of TESSA-RepCap5, TESSA-RepCap6, TESSA-RepCap8 and TESSA-RepCap9, and in each case, the AAV cap sequences were shown to be maintained in TESSA (Supplementary Fig. 7c–f). We have also confirmed the transgene expression cassette (CMV promoter and EGFP) within the rAAV genome to be stable at equal copy numbers relative to the adenovirus hexon gene up to the sixth passage in TESSA-AAV by qPCR (Supplementary Fig. 8a). Furthermore, the stability of the rAAV ITRs within the TESSA-AAV was validated by Illumina HiSeq next-generation sequencing (NGS) (Supplementary Fig. 8b). As expected, while the read depth is reduced in both ITR regions and consistent with the previous reports[41], we did not observe indels or SNPs within the AAV ITR regions from our sampling after filtering of reads with a Phred quality score of <20. Growing adenovirus achieves at least a 200-fold amplification per passage, and stability over six passages is sufficient to manufacture ~$6 × 10^{13}$ total adenoviral infectious units. Based on the data in Fig. 2a, this adenoviral material would be enough to produce $2 × 10^{17}$ rAAV vectors.

## Discussion

rAAV is the vector of choice for in vivo gene therapy for inherited genetic diseases, however, many treatments require high doses, presenting a significant hurdle for cost-effective manufacture[42–44]. Most rAAV manufacturing strategies currently use a three-plasmid 'helper-free' transfection approach[34,45]. However, this system is costly and challenging to scale up. We,

therefore, set out to use rational adenovirus engineering to develop a more efficient approach that completely avoids the use of transfection. We achieved this by inserting a tetracycline TetO site downstream of the adenovirus MLP TATA box and encoding TetR under its transcriptional control, creating the TESSA adenoviral vector in which the MLP is capable of self-repression. This approach avoids previously reported transcriptional leakage from TetR regulated systems[14,46] and achieves almost complete self-repression of adenovirus production. This is likely due to the dynamic linking of MLP activity to the level of the TetR transcripts i.e the more the MLP is active, the more it represses itself. Importantly, the expression of early adenovirus 'helper' genes, necessary for rAAV production, is maintained in TESSA[47]. We further extended the TESSA system by encoding both AAV rep and cap. This is the first demonstration of a stable adenovirus containing both of these genes and required that the rep gene (known to be toxic to adenovirus[22,27]) had no promoter and the p40 promoter region, also known to be a potent inhibitor of adenovirus replication, was scrambled by synonymous codon exchange. Surprisingly, despite attempts to limit the expression of Rep52/40 by ablation of the p19 TATA element, we found these proteins to be expressed at significant levels from TESSA. It, therefore, remains unclear whether ablation to the p19 TATA element assisted in relieving inhibition of adenovirus replication.

We also encoded the rAAV genome within a second TESSA virus, which when combined with TESSA-RepCap, created the TESSA2.0 system. We specifically avoided the use of HEK293T cells in this study to ensure regulatory compliance for clinical applications. The combination of these two adenoviral vectors for the production of rAAV2 gave a significantly greater yield of infectious rAAV2 than could be achieved using HF systems, reflecting both increased particle number and improved specific infectivity. The increased infectivity of rAAV produced using helper adenovirus compared to the plasmid transfection method has been reported previously[48,49]. Whilst the precise mechanism for this improvement in rAAV potency remains to be defined, our studies showed a fivefold increase in cellular uptake of TESSA-derived rAAV2 compared to HF-derived vectors, suggesting improved cell attachment and/or entry. We hypothesise that co-ordinated temporal expression of the adenoviral early genes and AAV Rep and Cap VP subunits from TESSA acts concertedly for efficient assembly of correct and infectious viral particles.

Although significant improvements in rAAV yield and quality have been reported before using helper adenoviruses[5,49,50], the TESSA system crucially produces extraordinarily low levels of adenoviral contamination (typically 2–15 TESSA particles per mL of total cell lysate and no wild-type adenovirus). Unlike other adenoviral helpers the TESSA virus is both E1/E3 deleted and MLP self-repressing, therefore any contaminating adenovirus is attenuated via two independent mechanisms, providing significant safety benefits. Existing affinity-based approaches to rAAV purification should be capable of removing such low levels of adenoviral particles. Indeed, purified rAAV stocks produced using the TESSA systems were routinely assessed and have been shown to be free of contaminating adenoviruses by the TCID50 assay (Supplementary Fig. 8c).

We chose to investigate both the TESSA2.0 adenoviral platform and its output AAV primarily for rAAV2, a serotype confounded by lower manufacturing yield compared to other naturally occurring AAV serotypes[51–53], although TESSA2.0-derived rAAV6, rAAV8 and rAAV9 also showed improved infectivity by 60-fold, 7-fold and 25-fold, respectively, compared to HF-derived reference stocks.

Having established several advantages of the TESSA2.0 helper system we then sought to propagate rAAV using rAAV stock as an input material alongside a single TESSA-RepCap2 vector. This system yields high titres of rAAV2, similar to the TESSA2.0 system, reducing manufacturing complexity. This strategy has the additional benefit of reduced reverse packaging compared to both TESSA2.0 and HF and raises the intriguing possibility that the quality of rAAV preparations may be improved by propagation[54]. Finally, the requirement that the rAAV efficiently infect the manufacturing cell line, in this case HEK293, led us to develop a capsid exchange approach. In this system the rAAV genome of interest is packaged in the AAV2 capsid to allow efficient infection of HEK293 alongside a TESSA virus encoding AAV rep and an alternative capsid. We demonstrated that this approach leads to high yields for rAAV5, rAAV6, rAAV8 and rAAV9 and can likely be used for any serotype. However, it can be expected that low levels of contaminant rAAV2 are present in the capsid exchanged preparations (such as rAAV5, 6, 8 and 9). Risks associated with these rAAV2 contaminants remain to be determined and their levels will likely be dependent on the quality of the rAAV2 seed-stock used for vector propagation. Specifically, as we observed $>3 \times 10^5$ fold amplification of rAAV5 per rAAV2 vector, this represented rAAV2 input contaminants to be present at a maximum of 0.003% at the pre-purification stage.

Currently the main strategies for rAAV manufacture are the HF and baculoviral expression systems, the latter producing rAAV in Sf9 insect cells. While high productivity comparable to the TESSA system has been reported from the baculoviral expression systems ($>1 \times 10^5$ GC/cell AAV2), there remain concerns over the potency of rAAV particles produced using these approaches[55,56].

In terms of commercial cost-effectiveness, the TESSA technologies provide enticing opportunities to produce large amounts of clinical-grade AAV of various serotypes, more efficiently and with less contamination by helper viruses than previous approaches have offered. This requires the TESSA adenoviral helper vectors to be produced to GMP at a significant scale. Previous studies have already shown the cost-effective scalability of adenovirus vectors up to 500 L and the global role out of adenoviral COVID-19 vaccine products further demonstrates that large quantities can be produced relatively easily in many different centres[57,58]. The approach we coin 'TESSA2.0' requires two such adenovirus vectors to be produced, whereas in the latter sections of our work, we describe the use of rAAV and TESSA-RepCap variants for rAAV manufacture by propagation instead of using two TESSA adenoviral vectors. This is a simple approach and may additionally present a scientific solution to the reduction of reverse packaged DNA in rAAV particles in comparison to common manufacturing approaches, giving a higher quality product. For the manufacture of difficult serotypes that do not infect packaging cells efficiently, we introduce the 'capsid-exchange' strategy which simply requires co-infection of rAAV2 with a TESSA-RepCap encoding the desired cap gene. Whilst the manufacture of rAAV2 seed materials required for both the 'propagation' and 'capsid-exchange' methods will add additional costs to these approaches, the potential to efficiently produce challenging AAV serotypes could be invaluable and warrants further investigation in terms of both commercial applicability and viability.

We believe the TESSA system offers a superior and scalable strategy for the manufacture of rAAV. The quality of individual AAV particles is similar to those obtained with regular adenovirus helper systems, however, the yields are appreciably higher, possibly reflecting no depletion of cellular resources for the production of adenovirus structural proteins. The ability to generate AAV particles demonstrating improved infectivity for a range of serotypes could be of particular importance to improve

the economic viability of rare disease treatments and allow decreased doses which may also reduce toxicity and enlarge the therapeutic window. Given the potential toxicities of high-dose AAV therapy, this could afford significant safety benefits. Finally, the TESSA system reduces adenovirus contamination to trace levels in crude lysate, and even those contaminants are doubly attenuated due to both E1 and TESSA modification. TESSA therefore has the potential to provide a scalable and cost-effective solution for the manufacture of AAV vectors for safe and efficient gene transfer.

## Methods

**Expression plasmid and viral vector generation.** Plasmids expressing EGFP transcribed from the MLP were constructed. Ad5 wildtype-MLP, MLP-TetO1a, MLP-TetO1b or MLP-TetO2 sequences were synthesised in a pUC57 plasmid backbone (Genewiz, NJ, USA) and used as a template for PCR with forward primer 5′-GAA-CATTTCTCTGTCGACCAACTAGTCGCCCTCTTCGGCATCAAG-3′ and reverse 5′-GGCCCTCGCAGACAGCGATGCGGAAGAGAG-3′, to generate DNA fragments with flanking homology sequences corresponding to the SalI restriction site position of plasmid OG186 pSF-pA-PromMCS (OXGENE, Oxford, UK) and the first-leader sequence of the adenovirus tripartite leader (TPL) (accession no: AY339865). A second DNA fragment encoding the EGFP reporter fused to the Ad5 TPL I-II-III was extracted by PCR from plasmid pSU109 using forward primer 5′-TCTCTTCCGCATCGCTGTCTGCGAGGGCC-3′ and reverse 5′-AGCTGAAGG-TACGCTGTATC-3′. MLP-TetO PCR fragments were cloned into a SalI and NheI linearised OG186 with the Ad-TPL-EGFP fragment by Gibson DNA assembly (New England Biolabs, MA, USA) to generate plasmids pMLPwt-EGFP, pMLP-TetO1a-EGFP, pMLP-TetO1b-EGFP, or pMLP-TetO2-EGFP. Plasmid OG268 containing a E1/E3-deleted Ad5 genome (OXGENE, Oxford, UK) was engineered to encode MLP-TetO1a, MLP-TetO1b, MLP-TetO2 sequences as follows. DNA sequences encoding the wildtype Ad5 MLP were extracted from plasmid OG617 (OXGENE, Oxford, UK), harbouring the full-length genome of wildtype Ad5, by PCR using forward primer 5′-CACTGACTGACTGATACAATCGATGAAGAAACGGTTTCCG-3′ and reverse 5′-CTGCGGATCCAGAAATCGATATCGATGCCGAAGGGGGCGTGGTC-3′ and cloned into ClaI linearised plasmid OG10 (OXGENE, Oxford, UK) Gibson DNA assembly (New England Biolabs, MA, USA) to create P4481. P4481 was further modified to add extended DNA sequences mapping to the E2B region of Ad5. E2B sequence was extracted by PCR using forward primer 5′-GCGCAGTACTGG-CAGCGGTGCACGGGCTGTACATCCTGCACGAG-3′ and reverse 5′-CGCTGTA TCTCAGTCAGTCAAGCTAGCGCAGCAGCCGCCGCGCC-3′ from OG268 and cloned into BsrGI and NheI linearised P4481 to create pSU43. DNA fragments corresponding to MLP-TetO1a, MLP-TetO1b, or MLP-TetO2 promoters were extracted from pUC57 plasmid backbone with PciI and DraIII and cloned into pSU43 linearised with the same enzyme to generate the MLP-TetO1a, MLP-TetO1b, and MLP-TetO2 shuttle plasmids. Sequence fragments encompassing Ad5 E2B with MLP-TetO1a, MLP-TetO1b, or MLP-TetO2 were extracted from the MLP shuttle plasmids using SphI and NheI and cloned into Bstz17I and XbaI linearised OG268 by Gibson DNA assembly (New England Biolabs, MA, USA) for exchange of the WT-MLP to create pSU208 (Ad5-MLP-TetO1a), pSU196 (Ad5-MLP-TetO1b), and pSU233 (Ad5-MLP-TetO2). For the construction of Ad5-MLP-TetO1b-SA-TetR (TESSA, pSU390) plasmid pSU196 was modified to insert a TetR coding sequence into the E3-deleted region. pSU79, an Ad5 E3-deleted-region shuttle plasmid was modified to insert a small multiple-cloning-site (MCS) by oligonucleotide annealing with forward primer 5′-CTTAAAATCAGTTAGCAAATTTATGCATGTCGACTAC GCCTCGAGGAGTAATCATTACGGGGGTC-3′ and reverse 5′-TATTAGTTAAAG GGAATAAGATCGCGACCTAG GGATCGTCGACGGTCGCACCTGAGGTGAC-GACTACCACATTTGTAGAG-3′ and cloned into pSU79 plasmid linearised with SalI to create pSU149. DNA fragments coding for the TetR was amplified by PCR using forward primer 5′-CTTAAAATCAGTTAGCAAATTTATGCATGCAGGAG-GAGGTACCCACCATGTCGCGCCTGGACAAAAG-3′ and reverse 5′-GAATAA-GATCGCGACCTAGGATAGCCGTCGACCTTTAAAAAACCTCCCACAC-3′ from template plasmid OG4156 (OXGENE, Oxford, UK) and cloned into pSU149 linearised with SalI enzyme to create pSU375. DNA fragments encoding TetR with flanking adenovirus DNA were extracted from pSU375 with enzymes SpeI and AscI and cloned into pSU196 linearised with the same enzymes to generate pSU382. Subsequently, DNA fragments corresponding to the L4 region of Ad5 were extracted from plasmid OG268 and ligated into NdeI linearised pSU382 to generate Ad5-MLP-TetO1b-SA-TetR (TESSA, pSU390). pSU390 or OG268 was further modified to insert the recombinant AAV2 genome, with a CMV-EGFP expression cassette, into the E1-deleted region. AAV2-CMV-EGFP fragment was excised from plasmid P3322 (OXGENE, Oxford, UK) using enzymes AsiSI and PacI and ligated into PacI linearised pSU390 and OG268 to generate plasmid pSU468 (TESSA-AAV) or pSU467 (Ad5-AAV), respectively. pAAV-CMV plasmid (Clontech, CA, USA) was modified to encode an EGFP coding sequence in the MCS to generate the pAAV-CMV-EGFP. For the construction of plasmid encoding the AAV transfer vector pscAAV-hFIX, DNA sequences corresponding to nucleotide 1 to 2362 of scAAV LP1-hFIXco[38] were synthesised as two gene fragments into pUC57 plasmids (Genewiz, NJ, USA). DNA fragments corresponding to the left and right section of LP1-

hFIXco were extracted from the pUC57 plasmids using restriction enzymes AflIII with XbaI, and BpuEI with PacI, respectively, and assembled into plasmid OG10 (OXGENE, Oxford, UK) linearised with AsiSI and SbfI using Gibson Assembly (New England Biolabs, MA, USA) to generate pSU878. For the construction of AAV transfer plasmid pAAV-hFIX, DNA fragment corresponding to the right AAV-ITR was extracted from pAAV-CMV-EGFP using MfeI and PacI and inserted into pSU878 digested with same restriction enzymes to generate pSU907. For the construction of TESSA-RepCap2, a 2.3Kb DNA fragment corresponding to AAV2 Rep78/68, wherein the p19 promoter and p40 inhibitory sequence was scrambled by synonymous codon exchange, was extracted from Ad5-E1-shuttle plasmid pSU633 using PacI restriction enzyme and inserted into pSU390 linearized with PacI to generate TESSA-Rep78/68 (pSU1091). Subsequently, a DNA fragment from Ad5-E1-deleted shuttle plasmid (pSU622) encoding the AAV Cap2 nucleotide sequence 1693 to 4220 from pRepCap2 (accession: AF369963.1), fused to the +1 transcriptional start site of the CMV promoter, was extracted by PacI restriction digest and inserted into AsiSI linearised pSU1209 (TESSA-RepCap2). For construction of TESSA-RepCap5, TESSA-RepCap6 and TESSA-RepCap8, plasmid (pSU1114) encoding AAV5 Cap encoding AAV5 Cap corresponding to nucleotide sequence 2210 to 4382 of the AAV5 genome (accession: AF085716.1), plasmid (R1149) encoding AAV6 Cap corresponding to nucleotide sequence 2210 to 4382 of the AAV6 genome (accession: AF028704.1), and plasmid (R1231) encoding AAV8 Cap corresponding to nucleotide sequence 2121 to 4338 of the AAV8 genome (accession: AF513852.1) under transcription control of a minimal CMV promoter was used as a PCR template for amplification using forward primer 5′-GTTGGCGTTTTATTAT-TATAGTCAGCTGACGGCGATTAAAAAAAACCTCCCACACCTCCCCCTG AACC-3′ and reverse primer 5′ CCCATCGATGGCGGCCGCCCCAGCGATTAA-GATCGATCTGTCGACCAACTACCCCGGGAAC-3′. The amplified AAV5, -6 and -8 cap sequences were inserted into AsiSI linearised plasmid pSU1091 using Gibson Assembly (New England Biolabs, MA, USA) to generate TESSA-RepCap5 (pSU1257), RepCap6 (R1236), TESSA-RepCap8 (R1238), respectively. For the construction of TESSA-RepCap9, and plasmid (R497) encoding AAV9 Cap corresponding to nucleotide sequence 1 to 2211 of the AAV9 hu.14 capsid protein VP1 (cap) gene (accession: AY530579.1) under transcriptional control of a CMV promoter was used as a PCR template for amplification using forward primer 5′-GTTGGCG TTTTATTATTATAGTCAGCTGACGGCGATTAAAAAAAACCTCCCACACCT CCCCCTGAACC-3′ and reverse primer 5′-GGTTTTTTTTAATTAACCCATC-GATGGCGGCCGCCCCAGCGATTAAGATCTAGTAATCAATCAGGGGTCA TTAG-3′. The amplified AAV9 cap sequences were inserted into AsiSI linearised plasmid pSU1091 using Gibson Assembly (New England Biolabs, MA, USA) to generate TESSA-RepCap9 (pSU1249).

**Cell culture.** HEK293 cells (293AD, Cell Biolabs, CA, USA) and Flp-In T-REx 293 cells (ThermoFisher Scientific, MA, USA) were cultured in Dulbecco's Modified Eagle Medium (DMEM; Sigma-Aldrich, MO, USA) supplemented with 10% (v/v) heat-inactivated foetal bovine serum (FBS; Gibco, MA, USA) and maintained at 5% $CO_2$, 37 °C, and 95% humidity. U-87 MG (ATCC, VA, USA) and HeLa RC32 (ATCC, VA, USA) were cultured in DMEM supplemented with 10% (v/v) heat-inactivated foetal bovine serum and maintained at 5% $CO_2$, 37 °C, and 95% humidity. Suspension HEK293 cells (OXGENE, Oxford, UK) were cultured in BalanCD (Irvine Scientific, CA, USA) supplemented with 4 mM Ultraglutamine I (Lonza, Basel, Switzerland). Cultures were maintained at 125 RPM, 37 °C, 8% $CO_2$ and 85% humidity.

**Adenovirus generation and titration.** All recombinant adenoviral vectors were initially recovered in HEK293 cells from plasmid DNA encoding each viral genome. Plasmids (20 μg) were linearised with SwaI restriction enzymes to release virus ITRs from the bacterial plasmid backbone and purified using a genomic Purelink DNA extraction kit (Invitrogen, CA, USA). HEK293 cells were seeded in T25 tissue culture flasks, at a density of $2 \times 10^6$ cells per flask, for 24 h before transfection. Each flask was transfected with 2.5 μg of linearised DNA using Lipofectamine 2000 (Invitrogen, CA, USA) according to the manufacturer's protocol. Transfection media were exchanged with fresh DMEM containing 2% FBS (supplemented with doxycycline 0.5 μg/mL or DMSO) after 4 h. Recombinant viruses were harvested from growth media ~12 days post-transfection upon observation of full CPE. Adenovirus vectors were further propagated in HEK293 cells and harvested by three rounds of freeze-thaw at day 3 post-infection. Cellular debris were pelleted by centrifugation and supernatant was passed through a 0.2 μm filter. For large-scale virus amplification and purification, HEK293 cells were seeded in Corning HYPERFlask (Sigma-Aldrich, MO, USA) for 48 h so that they are ~95% confluent before infection. TESSA-AAV or Ad5-AAV viral stock was used to infect each HYPERFlask at an MOI of 3 for 3 days. For TESSA-RepCap2, -5, -6, -8 and -9, an MOI of 15 was used for inoculation of HYPERFlasks. In addition, for the production of TESSA vectors, cell cultures were supplemented with doxycycline 0.5 μg/mL. Cells were harvested upon display of full CPE and the virus was released by three rounds of freeze-thaw. Viruses were purified by CsCl gradient banding, with Benzonase 250 U/mL (Sigma-Aldrich, MO, USA) added after the first round to degrade free DNA.

**TCID50 assay for determining infectious adenovirus.** HEK293 cells were seeded in 96-well tissue culture plates at a density of $1 \times 10^4$ cells per well for 24 h. Eight 10-fold serial dilutions of each adenovirus stock were made in DMEM containing 2% FBS at a total volume of 1.2 mL. For quantification of TESSA-based vectors,

DMEM was additionally supplemented with doxycycline 0.5 μg/mL. Ten replicates of each diluted sample ($1 \times 10^{-5}$ to $1 \times 10^{-12}$) were added at a volume of 100 μL per well on each plate. 100 μL of DMEM containing 2% FBS were added to the final two columns as a negative control. Plates were monitored over 12 days for the presence for viral plaques under a brightfield microscope. For TESSA-AAV and Ad5-AAV encoding the EGFP reporter, expression of EGFP observed under fluorescence microscopy was used as a proxy for infected cells. For determining infectious adenovirus contaminants in rAAV preparations generated from TESSA, eight 10-fold serial dilutions ($1 \times 10^{-1}$ to $1 \times 10^{-8}$) of the samples were made in DMEM containing 2% FBS and doxycycline 0.5 μg/mL. Ten replicates of each diluted sample were added at a volume of 100 μL per well on each plate and wells were monitored for the presence of viral plaques over 12 days. Infectious adenovirus was determined as TCID50 per mL using the KÄRBER-SPEARMAN statistical method[59]. For an MOI equal to one, cells were infected with 1 TCID50 unit per cell.

**Plasmid transfections for EGFP expression**. HEK293 cells were seeded for 24 h in 48-well tissue culture plates ($6.5 \times 10^4$ cells per well) before transfection with 0.75 μg per well of plasmid DNA using Lipofectamine 2000 according to the manufacturer's protocol. For co-transfections, plasmids pCMV-TetR and pMLP-TetO were used at a 1:1 ratio of DNA mass. Transfection mixture diluted in Opti-MEM (Gibco, MA, USA) were exchanged with DMEM containing 10% FBS, and supplemented with doxycycline 0.5 μg/mL (unless specified) or DMSO, at 4 hpt.

**rAAV vector production using adenovirus and HF system**. HF rAAV production was carried out according to Takara AAVpro Helper Free System (Clontech, CA, USA). Unless stated otherwise, rAAV vectors were produced in adherent HEK293 cells. HEK293 cells were seeded in 48-well tissue culture plates, at a density of $9 \times 10^4$ cells per well, for 24 h prior to production. For production via the HF method, each well was transfected with 0.75 μg of plasmid DNA, containing pHelper, pRepCap (Cap2, -5, -6, -8 or -9) and the rAAV transfer vector (pAAV-CMV-EGFP, pAA-hFIX, or pscAAV-hFIX) diluted with Opti-MEM (Gibco, MA, USA) at a DNA mass ratio of 1:1:1, and complexed using linear PEI 25kDA (Polysciences) at a 1:3 DNA to PEI mass ratio. For rAAV production using the TESSA2.0 system, HEK293 cells were co-infected (at the indicated vector dose) with TESSA-RepCap and TESSA-AAV diluted in DMEM containing 2% FBS. For propagation of rAAV using rAAV vectors, HEK293 cells were co-infected with rAAV2 vectors at 50 GC/cell with TESSA-RepCap (Cap2, -5, -6, or -9) at the indicated vector dose and diluted in DMEM containing 2% FBS. Transfection and infection media were exchanged with fresh DMEM containing 2% FBS (supplemented with doxycycline 0.5 μg/mL or DMSO as indicated) at 6 h post-treatment. To determine the efficiency of the DNase reaction in degrading free-DNA, a HF production control was included with each experiment, wherein the stuffer plasmid pUC19 was used in place of pRepCap. Similarly, for production control using the TESSA2.0 system, HEK293 cells were only infected with TESSA-AAV. rAAV vectors were harvested via three rounds of freeze-thaw of cells suspended in growth media. Cells were pelleted by centrifugation at 3000 g for 20 min and the supernatant was harvested for vector analysis. For rAAV production in suspension HEK293 cells, 25 mL of cell suspension were seeded in non-baffled E125 Erlenmeyer shake flasks at $1.5 \times 10^6$ cells per mL for 4 h before production treatment. Cells were transfected with the HF plasmids (pHelper, pRepCap (Cap2, -5, -6, -8 or -9) and pAAV-CMV-EGFP used at a DNA mass ratio of 1:1:1) at 1 μg per mL of cell suspension and complexed using PEIpro (Polyplus-transfection, NY, USA) at a 1:2 DNA to PEI mass ratio. For rAAV production via TESSA2.0 approach, suspension HEK293 cells were infected with TESSA-AAV and TESSA-RepCap (2, 6, 8 and 9) used at an MOI of 25. rAAV vectors from suspension cell cultures were harvested at 72 hpt by addition of 10X chemical lysis buffer (10 mM Tris, 20 mM MgCl₂, 1% (v/v) Triton X-100, pH 7.5) containing Benzonase ($1.25 \times 10^{-5}$ units per cell) and shaking at 200 RPM for 2 h 37 °C. NaCl was added to the lysate for a final concentration of 500 mM and the suspension was incubated for a further 1 h at 37 °C and shaking at 200 RPM. Cells were pelleted by centrifugation at 3000 g for 20 min and the supernatant was filtered using a 0.2 μm PES filter. For rAAV2 production from 1 L stirred-tank bioreactors (DASGIP Parallel Bioreactor Systems; Eppendorf, Hamburg, Germany), suspension HEK293 cells seeded at a $1.5 \times 10^6$ per mL were co-infected with TESSA2.0 vectors (TESSA-AAV-EGFP and TESSA-RepCap2, each at an MOI of 25) or co-infected with rAAV2-EGFP particles (HF-derived) at 50 GC/cell with TESSA-RepCap2 (used at an MOI of 25). rAAV vectors were harvested by chemical lysis, as stated previously. Supernatant was filtered using 0.2 μm PES filter and concentrated 10X by tangential flow filtration (Hollow fiber filter modules; Repligen, MA, USA). rAAV vectors produced in suspension HEK293 cells were purified by affinity chromatography using POROS™ GoPure™ AAVX affinity column (ThermoFisher Scientific, MA, USA).

**Viral RNA extraction and cDNA synthesis**. Total RNA was extracted using RNeasy Mini Kit (Qiagen, Venlo, Netherlands). RNA eluent (5 μL) was used for cDNA synthesis in a reverse transcription reaction (RT) using SuperScript IV First-Strand Synthesis System (Invitrogen, CA, USA). Five microlitres of each RT reaction were used as a template for qPCR to quantify viral RNA expression.

**Quantification of viral genomes and gene expression using qPCR**. For quantification of total adenovirus genomes in HEK293 cells, total DNA was extracted from culture media and cellular lysates using Purelink genomic DNA miniprep kit (Invitrogen, CA, USA). Five microlitres of DNA eluent were used in qPCR reactions using TaqMan Fast Advanced Master Mix (Applied Biosystems, CA, USA) in a StepOnePlus Real-Time PCR System (Applied Biosystems, CA, USA). Primer sequences for targeting Ad5 fibre are forward 5′-TGGCTGTTAAAGGCAGTTTGG-3′, reverse 5′-GCACTC CATTTTCGTCAAATCTT-3′ and probe 5′-TCCAATATCTGGAACAGTTCAAA GTGCTCATCT-3′. Primer sequences for targeting Ad5 hexon are forward 5′-CACT-CATATTTCTTACATGCCCACTATT-3′, reverse 5′- GGCCTGTTGGGCATAGA TTG-3′ and TaqMan probe 5′-AGGAAGGTAAC TCACGAGAACTAATGGG CCA-3′. Primer sequences for targeting TetR are forward 5′-ATGAGGTGGGAATT-GAAGGAC-3′, reverse 5′-CAGCATTTCGATGGCAAGC-3′ and TaqMan probe 5′-AAGAATAAACGGGCGCTCCTAGACG-3′. Primer sequences targeting the EGFP are forward 5′-GAACCGCATCGAGCTGAA-3′, reverse 5′-TGCTTGTCGGCCAT-GATATAG-3′, and TaqMan probe 5′-ATCGACTTCAAGGAGGACGGCAAC-3′. Primer sequences for targeting AAV2 rep forward 5′-GGCCTCATACATCTC CTTCAAT-3′, reverse 5′- AGTCAGGCTCATAATCTTTCCC-3′ and TaqMan probe 5′- TCCAACTCGCGGTCCCAAATCAA-3′. Primer sequences for targeting AAV2 cap are forward 5′-CGACCCAAGAGACTCAACTTC-3′, reverse 5′-GAACCGTGCTGG-TAAGGTTAT-3′ and TaqMan probe 5′-AAAGAGGTCACGCAGAATGACGGT-3′. Primer sequences for targeting hFIX are forward 5′-GTGTTCCCTGATGTGGAC-TATG-3′, reverse 5′- ACCCTGGTGAAGTCATTGAAG-3′ and TaqMan probe 5′-AGGCTGAAACCATCCTGGACAACA -3′. Primer sequences for targeting E4Orf6 are forward 5′-CACTACGACCAACACGATCT-3′, reverse 5′-GATGATCCTCCAG-TATGGTAGC-3′, and TaqMan probe 5′-AGACGATCCCTACTGTACGGAGTGC-3′. Primer sequences for targeting DBP are forward 5′-GACAGCGAGGAAGAAAGA-GAA-3′, reverse 5′- CTCCTTTGCCATGCTTGATTAG-3′, and TaqMan probe 5′-CGCTACAAATGGTGGGTTTCAGCA-3′. Primer sequences for targeting the AmpR are forward 5′-GCTGAATGAAGCCATACCAAAC-3′, reverse 5′-CTAGAGTAAGTA GTTCGCCAGTTAAT-3′ and TaqMan probe 5′-TGACACCACGATGCCTGTA GCAAT-3′. Primer sequences for the Ad5 packaging signal are forward 5′-CACA TGTAAGCGACGGATGT-3′, reverse 5′-CTTACTCGGTTACGCCCAAAT-3′, and TaqMan probe 5′-TTGTCACTTCCTGTGTACACCGGC-3′. Primer sequences for targeting the CMV promoter are forward 5′-CATATATGGAGTTCCGCGTTACAT-3′, reverse 5′- CTATTGGCGTTACTATGGGAACATAC-3′, and TaqMan probe 5′-TGGCTGACCGCCCAACGACC-3′. PCR cycles were as follows: 95 °C 10 min; 40 times (95 °C 1 s, 60 °C 20 s).

For quantification of genome encapsulated AAV and adenovirus particles, 2 μL of viral samples harvested from cell lysates preparations were treated with 1U of TURBO DNase (ThermoFisher Scientific, MA, USA) in a 20 μL reaction for 2 h at 37 °C. TURBO DNase was heat-inactivated at 75 °C for 10 min. Five microlitres of samples diluted at 1 in 200 using nuclease-free water were used in the qPCR reaction to quantify encapsulated Ad5 using Ad5 hexon primers and probe, while EGFP or hFIX primers and probe were used to quantify encapsulated AAV genomes. Standard curves for qPCR analyses were generated using a gBLOCK gene fragment encoding the qPCR amplicon sequences, (Integrated DNA Technologies, IA, USA) and Ct values of PCR reaction used to calculate DNA copy number by extrapolation to the standard curves. Titres of total AAV vectors produced using TESSA2.0 (including TESSA2.0-AAV6, -AAV8 and -AAV9) and HF method were determined by subtraction of baseline levels from TESSA production control (HEK293 cells infected with TESSA-AAV only) and the HF production control (stuffer pUC19 transfected in place of pRepCap), respectively. rAAV productivity per cell was determined from total rAAV vectors produced by cell count at point of transfection or infection using TESSA vectors.

**rAAV transduction assay**. rAAV transduction was measured using a TCID50 assay in HEK293 or Hela RC32 cells. For quantification in HEK293 cells, 96-well tissue culture plates were seeded at a density of $1 \times 10^4$ cells per well for 24 h. Eight 10-fold serial dilutions of each rAAV crude lysate stock were made in DMEM containing 2% FBS at a total volume of 1.2 mL. Ten replicates of each diluted sample ($1 \times 10^{-2}$ to $1 \times 10^{-10}$) were added at a volume of 100 μL per well on each plate. Hundred microlitres of DMEM containing 2% FBS were added to the final two columns as a negative control. Plates were observed for the presence of EGFP expressing cells from each well 6 days post-infection using a fluorescence micro-scope (EVOS FL imaging system, ThermoFisher Scientific, MA, USA). For quantification in Hela RC32 cells, 96-well tissue culture plates were seeded at a density of $2 \times 10^4$ cells per well for 24 h. Purified stocks of rAAV2 -6, -8 and -9 encoding EGFP produced from TESSA2.0 or HF method (Vector Biolabs, PA, USA) were quantified by EGFP-specific qPCR and normalised amounts of DNA-resistant genomes were serially diluted in DMEM containing 2% FBS and wt Ad5 (used at an MOI of 50). Eight 10-fold serial dilutions of each rAAV stock were prepared in triplicate and ten replicates of each diluted sample ($1 \times 10^{-2}$ to $1 \times 10^{-10}$) were added at a volume of 100 μL per well on each of the triplicate plates. Plates were observed for the presence of EGFP expressing cells from each well after 48 h using a fluorescence microscope. Transducing units per mL were calculated using the KÄRBER-SPEARMAN statistical method[59].

**AAV cell internalisation assay**. HEK293 cells were seeded in 48-well tissue culture plates at $7.5 \times 10^4$ for 24 h. Cells were counted before infection and rAAV2 derived from TESSA2.0 or HF method were used at 50 GC/cell. At 6 hpi, cells were trypsinised and extensively washed for three rounds with PBS before extraction of

total DNA using a DNeasy Blood & Tissue Kits (Qiagen, Venlo, Netherlands). Viral genomes were quantified by qPCR.

**Flow cytometry analysis**. Cells were harvested for flow cytometry at the indicated timepoint post-transduction by trypsinisation. Flow cytometry was performed using an Attune NxT Flow Cytometer (ThermoFisher, MA, USA) or Accuri C6 flow cytometer (BD Biosciences, NJ, USA). Gating strategy for excluding dead cells and defining the EGFP negative population of HEK293 cells are shown in Supplementary Fig. 9a–c.

**Western blot analysis**. Detection of adenovirus capsid proteins by western blotting was carried out using Wes automated system (ProteinSimple, CA, USA) according to the manufacturer's instructions and sample analysis carried out with Compass for SW software (ProteinSimple, CA, USA). Adenovirus was harvested from the culture medium and HEK293 cell lysate 3 days post-infection by three rounds of freeze-thaw. Cell debris was pelleted by centrifugation and lysates were probed using polyclonal antibodies directed against adenovirus type 5 (ab6982, Abcam, Cambridge, UK) and used at 1 in 50 dilution. For detection of TetR using Wes automated system, cellular extracts were harvested using RIPA buffer and probed using Anti-TetR antibody (ab25845, Abcam, Cambridge, UK) at 1 in 50 dilution. For detection of AAV2 Rep, cellular extracts were harvested from HEK293 cells using RIPA buffer. AAV capsid proteins were extracted by three rounds of freeze and thaw of the cell suspension, and the cellular debris was pelleted by centrifugation at 3000 g for 20 min. Equal amounts (25 μL) of total cellular extracts were resolved in 10% SDS-PAGE for detection AAV Rep and Cap proteins. Proteins were transferred to a nitrocellulose membrane using iBlot 2 Dry Blotting System (ThermoFisher, MA, USA) and probed with anti-AAV2 Rep antibody used at a 1 in 200 dilution (61069, clone 303.9, Progen, Heidelberg, Germany) or anti-AAV VP1/VP2/VP3 monoclonal antibody used at 1 in 200 dilution (65158, clone B1, Progen, Heidelberg, Germany). Anti-GAPDH antibody (2118, Cell Signaling Technology, MA, USA) was used at 1 in 1000 dilution. Secondary antibody donkey anti-mouse, HRP (A16011, ThermoFisher, MA, USA) was used at a 1 in 5000 dilution. Secondary antibody goat anti-rabbit, HRP (31460, ThermoFisher, MA, USA) was used at a 1 in 5000 dilution. HRP detected with TMB substrate solution (Sigma-Aldrich, MO, USA). Membranes were imaged using a Gel Doc XR + System (Bio-Rad Laboratories, UK).

**AAV capsid quantification by ELISA**. Quantification of assembled rAAV2, rAAV6, rAAV8 and rAAV9 capsids were carried out using AAV2 (PRAAV2R), AAV6 (PRAAV6), AAV8 (PRAAV8) and AAV8 (PRAAV9) titration ELISA kits (Progen, Heidelberg, Germany), respectively, according to the manufacturer's instruction. Reaction plates were read using a FLUOstar Omega micoplate reader (BMG LABTECH, Ortenberg, Germany).

**Transmission electron microscopy**. Electron microscopy imaging of rAAV particles was carried out by the Dunn School Bioimaging Facility, University of Oxford. For TEM imaging of purified rAAV2 produced in bioreactor HEK293 suspension culture cells from the TESSA2.0 or HF transfection method, 10 μL of neat HF-derived AAV or TESSA-derived AAV diluted 1 in 5 in PBS, were further diluted 1 to 1 with water and applied to freshly glow discharged carbon formvar 300 mesh copper grids for 2 min, blotted with filter paper and stained with 2% uranyl acetate for 10 s. Grids were imaged in a FEI Tecnai 12 TEM at 120 kV using a Gatan OneView CMOS camera.

**NGS analysis**. TESSA-AAV was sequentially passaged in HEK293 cells and viral genomes were extracted from CsCl-purified particles. 3.7 μg of purified DNA ($\sim 1 \times 10^{11}$ vector genomes) was used for sequencing by NGS Illumina HiSeq 2×250 bp run (Genewiz, NJ, USA). Sequencing reads were aligned to the reference sequence and visualized using Integrative Genomics Viewer (IGV, Broad Institute, MA, USA) and sequencing depth was determined by SAMtools depth (Genome Research Limited, UK). Burrows-Wheeler Aligner (BWA) alignment files were filtered by BCFtools (Genome Research Limited, UK) using the multi-allelic calling model, removing SNPs and indels with a Phred score of <20.

**Statistical analysis**. Data presented as mean ± standard error of mean (SEM), unless otherwise stated. Significance evaluated using unpaired, two-tailed Student's $t$-test, unless otherwise stated, and denoted on the graphs as *$P \leq 0.05$, **$P \leq 0.01$, ***$P \leq 0.001$, ****$P \leq 0.0001$. All statistical analysis was done using Prism 8 for Windows (GraphPad Software, USA).

**Reporting summary**. Further information on research design is available in the Nature Research Reporting Summary linked to this article.

## Data availability
All data generated throughout this study are included in this article and its supplementary information file. Vector sequences have been made publicly available through a published patent (PCT/GB2018/052083). Plasmid DNA and viral vectors generated from this study are available from the corresponding author on reasonable request. Source data are provided with this paper.

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

## Acknowledgements

This work was supported by the Biotechnology and Biological Sciences Research Council (BBSRC, W.S.) and Innovate UK Knowledge Transfer Network (W.S.). study. The authors thank Dr. Errin Johnson (Dunn School Bioimaging Facility, University of Oxford) for the TEM assistance and Dr. Pavneet Singh (OXGENE) for assistance in NGS analysis.

## Author contributions

W.S., L.W.S. and R.C. conceived the study. W.S., L.W.S and R.C. provided conceptual guidance and experiment design. W.S. carried out most of the experiments. M.I.P. generated various adenoviral vectors used in this study and carried out passages of various vector stocks. J.M.K. supervised rAAV production from bioreactors. W.S., L.W.S. and R.C. wrote the manuscript. L.W.S., M.R.D. and R.C. provided mentorship and supervision.

## Competing interests

W.S., M.I.P., J.M.K. and R.C. are employees of OXGENE. W.S., L.W.S. and R.C. are inventors on intellectual property related to this work. OXGENE is a company pursuing the development of TESSA for the commercial manufacture of rAAV. W.S. and R.C. have filed a patent application on the subject matter of this manuscript (PCT/GB2018/052083). M.R.D. declares no competing interests.
