## [Peer Review File · Nature Communications]

Reviewers' Comments:

Reviewer #1:

Remarks to the Author:

In this work, Su and colleagues devised and applied strategies to produce recombinant AAV vectors using novel, genetically engineered and self-inactivating Adenovirus vectors. In one iteration, the latter were further engineered to also express the AAV rep and cap genes, allowing for efficient amplification of existing AAV vectors through co-infection/-transduction of permissive cells.

As a whole, I found this work interesting and relevant for the AAV vector field, since scalable means for vector production or even amplification remain a bottleneck. Hence, the current work is a welcome addition to the ever growing arsenal of such technologies and certainly offers intriguing benefits.

One aspect that I found particularly interesting but that requires additional experimentation is that particles produced using the TESSA protocol seem to be more infectious as compared to those manufactured using a standard helpervirus-free protocol. How do the authors explain this, what could be the underlying mechanism and biology?

Moreover, I am wondering about the scalability, costs and work load of the serotype switching strategy they report at the end. To my understanding, this strategy either requires a permissive cell line for a given AAV capsid variant (which may rarely be available in the case of synthetic, organ-/cell-specific variants) or initial packaging into AAV2, in order to transduce HEK293T cells or similar. My concern is that the latter could quickly become cumbersome and costly upon upscaling? Do the authors really envision that the work, time and costs invoked by this strategy are outweighed by its benefits in the end?

Also, have the authors carefully assessed the stability of the ITRs in the AAV vector-encoding Adenovirus? Do they see recombinations and deletions over time? If so, are these a concern?

Reviewer #2:

Remarks to the Author:

I appreciate the huge amount of time the authors have spent on this work. More importantly, I think the technology described in their manuscript is valuable for the field.

AAV manufacturing is increasingly a hurdle in the gene therapy field, for large market disease targets and for rare diseases involving larger vector doses. This manuscript which describes a self-attenuating adenovirus. TESSA is an innovative new system for AAV production. All the experiments provide evidence that their modified adenovirus helper improves AAV production compared with helper-free systems.

I would like to provide some critique which I hope will be of use:

Major

Line 57 - "good functionality" is a little subjective. It would be good if the authors précised what they mean by "good"

Line 482 - the authors say "the ability to generate AAV particles demonstrating improved infectivity for a range of serotypes could potentially allow lower therapeutic doses for many AAV therapies"

Line 64 and thereafter - good to mention that the authors have used 293 rather than 293T, thus experiments are clinically more relevant...

Line 95, 1019,1126,1129 - with ANOVA, please state the post-hoc test that was used

I think this is a point which is not emphasized or discussed enough by the authors. I appreciate it has been discussed in other papers and reviews, but some discussion of why AAV particles produced by Ad helper/TESSA afford more productive gene transfer than particles produced by plasmid/helper free systems would be of interest and value. For instance, huge amounts of vector have been injected into babies with SMA, and I think even more into children with XLMTM. 60x, 7x and 25x are impressive improvements for AAVs 6,8 & 9

Lines 185-195 - the authors describe quite a few modifications to regions of wild type AAV - is it

the case they did them all at once and rolled the dice? If so, this might be a useful sentence in discussion, about the fact that individual modifications were not assessed and that it isn't clear which ones helped. Or maybe they did? I appreciate they state "rational combinatorial approach" but I think that may be too slick a statement for "we cleverly made lots of changes all at once" Line 453-455 "existing..should be capable.." Can the authors run a check and see? It just needs running down an AAV-X column or similar

Line 1010 - It is informative if the authors explain, here that Oxgene uses TESSA technology. And perhaps mention that there are patents filed on TESSA and technology described in this paper

Minor

Line 250 and throughout: It is great the authors compared against commercial reference material, and gives confidence that the improvements that TESSA provides are likely to the whole field Many figures - the authors have been excellent in providing individual values. Is the shaded bar really necessary? Or the SEM? With raw data points, the two are rendered (sic) unnecessary Lines 260 & 1075 - "verses" should be "versus"

Figure 4 and others - the numbers for the scale bars are tiny. The numbers for the scale bars need their own scale bars

Minor editing comment on the usage of a colon i.e. "Fig 4:" I believe the journal style is "Fig. 4"

Line 401 - Although I imagine the contamination is low to zero, it would be useful if the authors mentioned the actual likelihood of an alternative serotype prep being contaminated with AAV2.

E.g. it might even be useful to have this stated in the paper, if asked by a regulatory agency.

Line 425 - "rAAV is the vector of choice for in vivo gene therapy". Maybe say "...in vivo gene therapy for inherited genetic diseases". I think this is a little sweeping. Oncolytic adenoviruses are probably better for solid tumors although the authors may disagree.

Page 50 - "7 logs" might be better stated as 1×10^7 as a log can be in any base. Anyways, "7 logs" sounds like a Canadian microbrewery.

POINT-BY-POINT RESPONSE FORM

Manuscript #: NCOMMS-21-25776

Manuscript title: Self-attenuating adenovirus enables propagation of recombinant adeno-associated virus for high manufacturing yield without contamination

Suggestion, Question, or Comment from Reviewer #1	Author’s Response	Change in the Manuscript
One aspect that I found particular interesting but that requires additional experimentation is that particles produced using the TESSA protocol seem to be more infectious as compared to those manufactured using a standard helpervirus-free protocol. How do the authors explain this, what could be the underlying mechanism and biology?	The phenomenon that AAV produced using helper adenovirus has greater specific infectivity than AAV produced using helper-free systems has been reported before. We believe the mechanism involved – most likely reflecting the involvement of adenovirus proteins in fine tuning the structure of the AAV particles (i.e. adenovirus proteins that are not essential for AAV production, and not included in the helper-free system) – is likely to be the same as we observed here using TESSA. This phenomenon was a major motivation for our development of TESSA, as it seemed likely that a contaminant-free adenovirus-based system would yield AAV with improved specific infectious activity. No group has yet identified precisely which steps of AAV infection are improved; our studies (Figure 2e) do indicate a 5-fold increase in cell uptake for AAV2 made using TESSA compared to helper-free, hence improved cellular uptake is likely to comprise an important part of the answer. Ultimately, we believe several steps of infection may be improved using AAV made with helper adenovirus, but the specific details require extensive experimentation to define and would likely be the subject of additional publications. We have re-emphasised that rAAV made using helper adenovirus is known to show greater infectious activity than the helper-free method.	Added to discussion (Line 455-460) “The increased infectivity of rAAV produced using helper adenovirus compared to the plasmid transfection method has been reported previously (49; 50). Whilst the precise mechanism for this improvement in rAAV potency remains to be defined, our studies showed a 5-fold increase in cellular uptake of TESSA-derived rAAV2 compared to helper-free vectors, suggesting improved cell attachment and/or entry.”
Moreover, I am wondering about the scalability, costs and workload of the serotype switching strategy they report at the end. To my understanding, this strategy either requires a permissive cell line for a given AAV capsid variant (which may rarely be available in the case of synthetic, organ-/cell-specific variants) or initial packaging into AAV2, in order to transduce HEK293T cells or similar. My concern is that the latter could quickly become cumbersome and costly upon upscaling? Do the authors really envision that the	The development of target-specific AAV (which by definition will not infect via common receptors) seems likely to bring significant manufacturing challenges, not usually met by existing packaging cells. Producing permissive recombinant cells is one option, but we believe in comparison the capsid switch approach is much more affordable and quicker. Detailed assessment of the commercial potential of this approach was beyond the scope of this work, nevertheless we wanted to introduce it as a novel scientific concept and believe it provides an interesting strategy for manufacture of the different AAV serotypes and as a mechanism to reduce ‘reverse-packaged’ DNA impurities.	As the reviewer comment primarily relates to the commercial viability of the approach, we did not feel it appropriate to modify what is essentially a research publication. However, we are happy to add additional discussion to the manuscript should the reviewer feel that it is required.

work, time and costs invoked by this strategy are outweighed by its benefits in the end?	As a rough estimate of required manufacturing scales, based on the observed productivity ($>1 \times 10^6$ GC/cell) that we were able to achieve for rAAV5 and rAAV8 using the capsid exchange method it can be estimated that $\sim 1.5 \times 10^{13}$ of rAAV2 would be required for inoculating 200L of HEK293 cell culture and sufficient for generation of $\sim 3 \times 10^{17}$ rAAV5 and -8 vector genomes. While the economic cost of generating rAAV2 seed stock remains to be determined, based on the yield ($\sim 1 \times 10^4$ GC/cell) of rAAV2 that we were able to achieve using the helper-free plasmid approach, a 1L cell culture production should be sufficient to generate $\sim 1.5 \times 10^{13}$ of rAAV2 vector. Fundamentally, we do agree with the reviewer that it warrants further investigation to understand commercial viability. However, the numbers above suggest that significant cost savings can be achieved relative to the plasmid process.	
Also, have the authors carefully assessed the stability of the ITRs in the AAV vector-encoding Adenovirus? Do they see recombinations and deletions over time? If so, are these a concern?	The concept of incorporating a rAAV genome into adenoviral vectors has been shown by others to be a viable approach (references below) and instability of the ITR sequence has not been reported in these publications. Consistently with these publications, we did not observe instability during the work presented in the manuscript. We have amended the manuscript for better clarification. References Sun, B., Y. T. Chen, A. Bird, F. Xu, Y. X. Hou, A. Amalfitano, and D. D. Koeberl. Packaging of an AAV vector encoding human acid alpha-glucosidase for gene therapy in glycogen storage disease type II with a modified hybrid adenovirus-AAV vector. Mol Ther 7. 467-477 (2003). Fisher, K. J., W. M. Kelley, J. F. Burda, and J. M. Wilson. A novel adenovirus-Adeno-associated virus hybrid vector that displays efficient rescue and delivery of the AAV genome. Hum Gene Ther 7. 2079-2087 (1996). Liu, X. L., K. R. Clark, and P. R. Johnson. Production of recombinant Adeno-associated virus vectors using a packaging cell line and a hybrid recombinant adenovirus. Gene Ther 6. 293-299 (1999). Gao, G.P., Qu, G., Faust, L.Z., Engdahl, R.K., Xiao, W., Hughes, J.V., Zoltick, P.W., and Wilson, J.M. High-titer Adeno associated viral vectors from a Rep-Cap cell line and hybrid shuttle virus. Hum Gene Ther 9. 2353-2362 (1998).	Added to discussion (Line 449-451) “During this study, no instability of the rAAV genome was observed by qPCR up to the sixth passage (data not shown) and consistent with previous publications”

Suggestion, Question, or Comment from Reviewer #2	Author's Response	Change in the Manuscript
Line 57 - "good functionality" is a little subjective. It would be good if the authors precised what they mean by "good"	“Good functionality” implies that basal transcription activity of the MLP was not significantly hindered by insertion of the TetO sequences. We have amended the manuscript for better clarification.	Line 56-57 changes “We first identified MLP regions that allow insertion of TetO sites whilst maintaining wildtype MLP basal transcriptional activity”.
Line 482 - the authors say "the ability to generate AAV particles demonstrating improved infectivity for a range of serotypes could potentially allow lower therapeutic doses for many AAV therapies"	We agree this is an important point, with exciting potential. However, we are unclear precisely what amendments the referee is requesting, but have ensured that the message conveyed is clearer in the revised manuscript.	Added to discussion (Line 503-506) “The ability to generate AAV particles demonstrating improved infectivity for a range of serotypes could be of particular importance to improve economic viability of rare disease treatments and allow decreased doses which may also reduce toxicity and enlarge the therapeutic window.”
Line 64 and thereafter - good to mention that the authors have used 293 rather than 293T, thus experiments are clinically more relevant...	This comment has now been acknowledged in the text.	Added to discussion (Line 451-452) “We also specifically avoided the use of HEK293T cells in this study to ensure regulatory compliance for clinical applications.
Line 95, 1019, 1126, 1129 - with ANOVA, please state the post-hoc test that was used I think this is a point which is not emphasized or discussed enough by the authors.	This comment has now been acknowledged in the text.	Added to discussion Line 97 - “analyzed by two-way ANOVA followed by Bonferroni post hoc test comparing TESSA-AAV (DMSO) versus TESSA-AAV (DOX), Ad5-AAV (DOX) and Ad5-AAV (DMSO)” Line 1062, 1169 - “analyzed by two-way ANOVA with Bonferroni post hoc testing”

I appreciate it has been discussed in other papers and reviews, but some discussion of why AAV particles produced by Ad helper/TESSA afford more productive gene transfer than particles produced by plasmid/helper free systems would be of interest and value. For instance, huge amounts of vector have been injected into babies with SMA, and I think even more into children with XLMTM. 60x, 7x and 25x are impressive improvements for AAVs 6, 8 & 9.	We agree the observed better quality preps seen with Ad helper compared to helper-free has been seen before, and presumably reflects involvement of other adenovirus proteins in refining the assembly of the AAV particles, which are not present in the helper-free system. However, we are unable to make specific mechanistic comments on the matter as the biological cause has yet to be elucidated, but is the subject of further work by our labs. That said we have added additional comments to the discussion based on the data we have available.	As addressed in our response to the first point of referee #1 We have amended the discussion to comment on this issue.
Lines 185-195 - the authors describe quite a few modifications to regions of wild type AAV - is it the case they did them all at once and rolled the dice? If so, this might be a useful sentence in discussion, about the fact that individual modifications were not assessed and that it isn't clear which ones helped. Or maybe they did? I appreciate they state "rational combinatorial approach" but I think that may be too slick a statement for "we cleverly made lots of changes all at once"	The generation of adenovirus vectors stably expressing the AAV Rep and Cap genes was a longstanding aim in our lab, and one which was extensively explored in the past by other research groups. Our rational combinatorial approach to enable stable propagation of AAV Rep within the adenovirus/TESSA genome was based on previous publications detailing the inhibitory effect of the Rep proteins and its coding sequence on adenovirus replication. Consistent with previous publications, our preliminary work also found that limiting expression of Rep78/68 and scrambling the p40-inhibitory sequence by synonymous codon exchange were both essential as we were unable to recovery viable viruses when only one of these changes were introduced. However, a surprising finding was that despite our pre-emptive attempts to limit expression of Rep52/40 by ablation of the p19 TATA element, we found these proteins to be expressed at significant levels from the TESSA virus (Supplementary Fig. 3g). We are therefore uncertain whether this change was essential to enable stable propagation of rep within the adenovirus and have revised the manuscript to reflect this fact.	Added to discussion (Line 441-447) “This is the first demonstration of a stable adenovirus containing both these genes and required that the rep gene (known to be toxic to adenovirus (22; 27)) had no promoter and the p40 promoter region, also known to be a potent inhibitor of adenovirus replication, was scrambled by synonymous codon exchange. Surprisingly, despite attempts to limit expression of Rep52/40 by ablation of the p19 TATA element, we found these proteins to be expressed at significant levels from TESSA. It therefore remains unclear whether ablation to the p19 TATA element assisted in relieving inhibition of adenovirus replication.”
Line 453-455 "existing..should be capable.." Can the authors run a check and see? It just needs running down an AAV-X column or similar	This point was discussed further in line 470-472: “Indeed, purified rAAV stocks produced using the TESSA systems were routinely assessed and have been shown to be free of contaminating adenoviruses by the TCID50 assay (not shown)”.	No changes added.

Line 1010 - It is informative if the authors explain, here that Oxgene uses TESSA technology. And perhaps mention that there are patents filed on TESSA and technology described in this paper	This comment has now been acknowledged in the text.	Added to conflict of interest - “W.S, L.W.S and R.C are inventors on intellectual property related to this work. OXGENE is a company pursuing the development of TESSA for the commercial manufacture of rAAV.”
Line 250 and throughout: It is great the authors compared against commercial reference material, and gives confidence that the improvements that TESSA provides are likely to the whole field	We share the reviewer’s view and are grateful that he/she flagged it up.	No changes made
Many figures - the authors have been excellent in providing individual values. Is the shaded bar really necessary? Or the SEM? With raw data points, the two are rendered (sic) unnecessary	This an interesting point – normally reviewers ask for more analysis not less. In this case, showing the mean and the error bars is arguably superfluous but only because the raw data points are closely clustered together. If the data points were dispersed, showing these parameters would be essential. Hence on balance we hope the reviewer will agree to let us keep them.	No changes made
Lines 260 & 1075 - "verses" should be "versus"	This comment has now been acknowledged in the text.	Line changes Lines 262 & 1118 – “versus”
Figure 4 and others - the numbers for the scale bars are tiny. The numbers for the scale bars need their own scale bars	This comment has now been acknowledged in the text.	Numbers in scale bars have been enlarged.
Minor editing comment on the usage of a colon i.e. "Fig 4:" I believe the journal style is "Fig. 4"	We have checked all references to the figures in the manuscript and they appear to be in compliance with the journal style. Please let us know if further amendments are required.	No changes made
Line 401 - Although I imagine the contamination is low to zero, it would be useful if the authors mentioned the actual likelihood of an alternative serotype prep being contaminated with AAV2. E.g. it might even be useful to have this stated in the paper, if asked by a regulatory agency.	This comment has now been acknowledged in the text.	Added to discussion (Line 488-494) “However, it can be expected that low levels of contaminant rAAV2 are present in the capsid exchanged preparations (such as rAAV5, 6, 8 and -9). Risks associated with these rAAV2 contaminants remains to be determined

		and their levels will likely be dependent on the quality of the rAAV2 seed-stock used for vector propagation. Specifically, as we observed $>3 \times 10^5$ fold amplification of rAAV5 per rAAV2 vector, this represented rAAV2 input contaminants to be present at a maximum of 0.003% at the pre-purification stage”
Line 425 - "rAAV is the vector of choice for in vivo gene therapy". Maybe say "...in vivo gene therapy for inherited genetic diseases". I think this is a little sweeping. Oncolytic adenoviruses are probably better for solid tumors although the authors may disagree.	This comment has now been acknowledged in the text.	Line 427 changed “rAAV is the vector of choice for in vivo gene therapy for inherited genetic diseases”
Page 50 - "7 logs" might be better stated as 1×10^7 as a log can be in any base. Anyways, "7 logs" sounds like a Canadian microbrewery.	This comment has now been acknowledged in the text.	Line 50 changed to “Adenoviral contamination levels were reduced by 1×10^7 fold”

Reviewers' Comments:

Reviewer #1:

Remarks to the Author:

I thank the authors for addressing my comments and concerns.

However, I do not understand how they presumably monitored ITR stability by qPCR? How did the assay work? Why did they not sequence the ITRs?

Reviewer #2:

Remarks to the Author:

I would like to thank the authors for addressing my comments (at least the ones that were reasonable) and would like to congratulate them on an detailing an exciting technology.

POINT-BY-POINT RESPONSE FORM

Manuscript #: NCOMMS-21-25776

Manuscript title: Self-attenuating adenovirus enables propagation of recombinant adeno-associated virus for high manufacturing yield without contamination

Suggestion, Question, or Comment from Reviewer #1	Author’s Response	Change in the Manuscript
However, I do not understand how they presumably monitored ITR stability by qPCR? How did the assay work? Why did they not sequence the ITRs?	Our qPCR assay is designed to confirm stability of the transgene expression cassette within the rAAV genome that is encoded in the TESSA vectors. In this assay, TESSA-AAV-EGFP vectors are serially passaged in HEK293 cells and viral genomes extracted from CsCl-purified particles for qPCR. The qPCR detection assay uses primers and probe directed against various regions of the transgene expression cassette within the rAAV genome and the adenovirus Hexon coding sequence to ensure an equal relative copy number between these DNA sequences. Stability assessment of the CMV and EGFP coding sequence within TESSA-AAV by qPCR is provided in Response figure 1 (below). In response to the Reviewer’s final query, Next Generation Sequencing (via Illumina HiSeq 2x250 bp run) was also carried out to confirm sequence fidelity and variations within the TESSA-AAV-EGFP vectors. Analysis of read depth and coverage against reference sequence is also provided in Response figure 2a and 2b. The vast majority of reads (98% out of 5.1 million reads) aligned to the reference sequence and alignment had sufficient depth and coverage to allow for further SNP and indel analysis. As expected, while the read depth is reduced in both ITR regions and consistent with previous reports (Petri et al 2014), we did not observe indels or SNPs within the AAV ITR regions from our sampling after filtering out of reads with a quality of <20 Phred. This work is part of a larger study in our stability assessment of TESSA vectors encoding different rAAV genomes/transgene cassettes, and rAAV vector stability at sequential propagation. Results of this work are expected to be the subject of additional publications We have amended the manuscript to clarify the qPCR assay used for stability assessment of the transgene expression cassette within the rAAV transfer genome and to outline the salient results of the NGS.	Changes in discussion (line 449-452): “Consistent with previous publications, the transgene expression cassette within the rAAV genome was shown to be stable in TESSA-AAV by qPCR and next-generation sequencing (NGS) up to the sixth passage (data not shown)”

	Petri, K., Fronza, R., Gabriel, R., Kappel, C., Nowrouzi, A., Linden, R.M., Henckaerts, E. and Schmidt, M. Comparative next-generation sequencing of adeno-associated virus inverted terminal repeats. Biotechniques 56. 269-273 (2014).	
The editorial team has been discussing the concerns from the previous round regarding scalability of this approach. Since as a journal we are interested in applied science with translational application, in addition to primary research papers, we ask that the concerns of Reviewer #1 from the previous round of review are addressed through open and honest discussion regarding the manufacturing challenges. Though we do not require experimental evidence of scalability, we are interested in how this would be achieved. ORIGINAL concerns from Reviewer 1 – “Moreover, I am wondering about the scalability, costs and workload of the serotype switching strategy they report at the end. To my understanding, this strategy either requires a permissive cell line for a given AAV capsid variant (which may rarely be available in the case of synthetic, organ-/cell-specific variants) or initial packaging into AAV2, in order to transduce HEK293T cells or similar. My concern is that the latter could quickly become cumbersome and costly upon upscaling? Do the authors really envision that the work, time and costs invoked by this strategy are outweighed by its benefits in the end?”	We respectfully suggest that the editorial team may have missed that this comment relates only to an offshoot of the technology. The major focus of the paper is about TESSA, and the scalability of that is not in doubt (see for example the scale-up data in Figure S4). This comment from Reviewer #1 relates to the ‘capsid switch’ technique (Fig.4 (g)-(j)), which is included as an intriguing but non-essential offshoot that should provide a means to apply TESSA technology to manufacture an emerging generation of target-specific AAV, rather than the conventional serotypes that currently dominate the field. Regarding the scalability of the ‘offshoot’ technology, the reviewer comments that ‘packaging into AAV2...could rapidly become cumbersome and costly upon upscaling’. We anticipate in most situations the transgene of interest will already be in clonable form for insertion into AAV2, and the novel capsid gene has to be inserted into TESSA-repcap. From there on the production is similar to that validated for TESSA in the early part of the paper, and our assessments of yield (see earlier response) indicate efficient production is likely to be feasible. Hence the only aspects of the capsid switch technology that seem likely to cause costs or delays are the two cloning steps, both of which can be completed by individual scientists in a matter of weeks, and production of the TESSA-repcap. Accordingly, we do not think this will have a significant effect on overall cost and efficiency, although we do agree with the referee that it would be good to see this put into practice at scale and we have expanded discussion on this point. As previously mentioned, as a rough estimate of required manufacturing scales, based on the observed productivity ($>1 \times 10^6$ GC/cell) that we were able to achieve for rAAV5 and rAAV8 using the capsid exchange method it can be estimated that $\sim 1.5 \times 10^{13}$ of rAAV2 would be required for inoculating 200L of HEK293 cell culture and sufficient for generation of $\sim 3 \times 10^{17}$ rAAV5 and -8 vector genomes. While the	Added to discussion (line 501-509) “The ability to manufacture large quantities of rAAV using the technologies described herein require that adenoviral vectors be produced at significant scale. Previous studies have demonstrated the scalability of such vectors up to 500 L and the global role out of adenoviral COVID-19 vaccine products further demonstrates that such quantities can be achieved (58; 59). In the latter sections of our work, we describe the use of rAAV and TESSA-RepCap variants for rAAV manufacture instead of using two TESSA adenoviral vectors. This approach may present a scientific solution to the reduction of reverse packaged DNA in rAAV particles in comparison to both the TESSA2.0 and plasmid-based methods, and warrants further investigation in terms of both commercial applicability and viability.”

	economic cost of generating rAAV2 seed stock remains to be determined, based on the yield (~1x10⁴ GC/cell) of rAAV2 that we were able to achieve using the helper-free plasmid approach, a 1L cell culture production should be sufficient to generate ~1.5x10¹³ of rAAV2 vector. Fundamentally, we do agree with the reviewer that it warrants further investigation to understand commercial viability. However, the numbers above suggest that significant cost savings can be achieved relative to the plasmid process, and the large-scale manufacture of adenoviral vectors has been extensively demonstrated by others, particularly during the development and deployment of adenoviral based COVID19 vaccines.	
NA	NA	Reference added: “58. Shen C.F. et al. Process optimization and scale-up for production of rabies vaccine live adenovirus vector (AdRG1.3). Vaccine 30, 300-306 (2012). 59. Voysey, M. et al. Safety and efficacy of the ChAdOx1 nCoV-19 vaccine (AZD1222) against SARS-CoV-2: an interim analysis of four randomised controlled trials in Brazil, South Africa, and the UK. Lancet 397, 99-111 (2021).” Acknowledgements added: “Dr. Pavneet Singh (OXGENE) for assistance in NGS analysis.”

Response figure 1: Assessment of TESSA-AAV genetic stability at serial passages by qPCR analysis. DNA extracted from purified TESSA-AAV vectors at the indicated passages were quantified by qPCR against the CMV promoter, EGFP, and Ad5 hexon. Data shows the copy number of CMV and EGFP DNA relative to Ad5 hexon and compared to DNA plasmid encoding TESSA-AAV (data as N=3; ns, analysed by two-way ANOVA with Bonferroni post hoc testing).

a**b**
Response figure 2: NGS sequencing of TESSA-AAV via Illumina HiSeq 2x250 bp run. TESSA-AAV was sequentially passaged in HEK293 cells and viral genomes were extracted from cesium chloride-purified particles. 3.7 μ g of purified DNA ($\sim 1 \times 10^{11}$ vector genomes) was used for sequencing. (a) Sequencing reads were aligned to the reference sequence and visualized using Integrative Genomics Viewer (IGV) and (b) sequencing depth was determined by SAMtools depth. Base position and sequencing depth of the AAV ITRs, transgene expression cassette, TetO, and TetR coding sequences within TESSA-AAV are highlighted.